# Single-cell analyses identify monocyte gene expression profiles that influence HIV-1 reservoir size in acutely treated cohorts

Eliminating latent HIV-1 is a major goal of AIDS research but host factors determining the size of these reservoirs are poorly understood. Here, we investigate the role of host gene expression on HIV-1 reservoir size during suppressive antiretroviral therapy (ART). Peripheral blood cells of fourteen males initiating ART during acute infection and demonstrating effective viral suppression but varying magnitudes of total HIV-1 DNA were characterized by single-cell RNA sequencing. Differential expression analysis demonstrates increased CD14+ monocyte activity in participants having undetectable HIV-1 reservoirs, with *IL1B* expression inversely associating with reservoir size. This is validated in another cohort of 38 males comprised of different ancestry and HIV-1 subtypes, and with intact proviral DNA assay (IPDA®) measurements. Modeling interactions show monocyte *IL1B* expression associates inversely with reservoir size at higher frequencies of central memory CD4+ T cells, linking monocyte *IL1B* expression to cell types known to be reservoirs for persistent HIV-1. Functional analyses reveal that IL1B activates NF-κB, thereby promoting productive HIV-1 infection while simultaneously suppressing viral spread, suggesting a natural latency reversing activity to deplete the reservoir in ART-treated individuals. Altogether, scRNA-seq analyses reveal that monocyte *IL1B* expression could decrease HIV-1 proviral reservoirs in individuals initiating ART during acute infection.

HIV-1 infection is effectively treated with antiretroviral therapy (ART). However, the persistence of stably integrated and replication-competent proviruses in the latent cell reservoir despite treatment prevents a cure[1]. ART suppresses plasma viremia below the limit of detection but viral replication rebounds within weeks of analytic treatment interruption (ATI) in the majority of individuals[2]. However, there is increasing evidence that the pool of latently infected cells harboring HIV proviruses[3] varies in size between individuals. Several studies have implicated resting memory CD4+ T cells and distinct memory CD4+ T cell compartments as the primary latent reservoirs in people living with HIV (PLWH) on ART[4–8]. Variation in reservoir size as determined by HIV DNA quantification in CD4+ T cells has also been observed in virally suppressed patients who initiated ART during acute

HIV infection[9,10]. This inter-host variation in CD4+ T cell-associated reservoir size is observed at various stages of acute infection and even after 24 weeks of ART[9,11]. Identifying host cellular factors that mark and influence the HIV reservoir size could help in understanding the mechanisms associated with HIV persistence and may reveal targets for achieving a functional cure.

The majority of previous findings linking the host transcriptome to latency have been limited to cell lines or models of infection, and ex vivo experiments with primary cells[12–14]. Recent studies in humans have focused on assessing CD4+ T cells and HIV persistence in the context of characterizing antigenicity, clonal expansion, and the whole transcriptomes of bulk or single cells harboring virus[15–19]. Here, we determined cellular immune profiles of the host in peripheral blood

✉ e-mail: rthomas@hivresearch.org

that correlate with cell-associated HIV DNA levels, an established marker of viral persistence, comparing extreme phenotypes of reservoir size[20]. These in vivo quantitative phenotypes of multiple participants enable unbiased approaches to interrogate all cell populations without ex vivo stimulation. A unique cohort of PLWH that initiated ART treatment during Fiebig stage III of acute infection was selected in order to minimize the effect of time-to-treatment as a confounder of reservoir size[21]. This was combined with the use of single-cell analytical approaches that are high throughput and not based on a priori knowledge to avoid specifically targeting the known latently infected T cell reservoir, and to enable broad screening for host variation most prominently associated with reservoir size.

Given the sustained size variation in cellular reservoirs during acute HIV infection (AHI) and post-ART initiation, we hypothesized that specific host genes might contribute to these differences between individuals and could be identified using transcriptomics. We have previously shown that a bulk RNA-seq approach applied to multiple sorted lymphocyte populations allowed us to identify protective gene signatures in response to HIV vaccination[22]. Here, we used a more sensitive next-generation sequencing (NGS) approach to identify differences in host transcriptomes from PLWH shown to harbor varying HIV DNA levels[21]. Additionally, we recently showed that transcriptomics studies conducted with AHI samples can be confounded by the presence of viremia[18,23]. In this study, we achieved broader scope and resolution using a droplet-based single-cell RNA-seq (scRNA-seq) platform with peripheral blood mononuclear cell (PBMC) samples from virally suppressed PLWH. This enabled the examination of gene expression in all cell types in peripheral blood, to test expression of every gene in the transcriptome from individual cells for associations with the size of the total cell-associated HIV reservoir in participants on ART. Our single-cell analyses identified monocyte gene expression profiles as having the strongest association with HIV reservoir size in two independent AHI cohorts. Specifically, higher *IL1B* expression in CD14+ monocytes was associated with smaller reservoirs in both studies. Moreover, we were able to confirm these findings with intact proviral DNA assay (IPDA®) measurements of intact HIV-1 proviruses. Functional in vitro data support an effect via IL1B-mediated activation of the NF-κB transcription factor family, which both stimulates HIV transcription and induces antiviral gene expression[24].

## Results

### Participant selection and study design

We screened 163 Thai participants enrolled in the RV254 AHI cohort with varying cell-associated HIV-1 DNA reservoir sizes to map the immune-microenvironment of PLWH. We further focused on performing integrated transcriptome-wide scRNA-seq, high parameter multidimensional flow cytometry, T-cell receptor (TCR) and B-cell receptor (BCR) sequencing from PBMC of 14 selected participants who had been on ART for 48 weeks. All individuals had initiated ART following HIV diagnosis during Fiebig stage III of AHI and were virally suppressed ( < 50 copies/ml) (Supplementary data file 1). We categorized the 14 participants with the most extreme reservoir size differences into "undetectable" ($n = 8$) versus "detectable" ($n = 6$) reservoir groups from a total of 28 participants at Fiebig stage III (Fig. 1a, b). These phenotypes were based on total cell-associated HIV DNA and confirmed using integrated HIV DNA levels in PBMC as measured by quantitative PCR[10,25]. Further, HIV DNA decay from week 0 at AHI showed that the phenotype categorizations were distinct (Fig. 1c). All 14 participants shared similar demographics and were Thai males infected with viral subtype CRF01_AE, as described previously[18]. Other than reservoir size there were no significant differences between the two groups (Fig. 1b, d). The workflow, including scRNA-seq, repertoire sequencing, and flow cytometry, performed on samples from all 14

participants is illustrated in Fig. 1e. Furthermore, PBMC from an additional 38 male participants with viral subtype B infections and African and European ancestry from the USA (ACTG A5354) were assessed 48 weeks after ART initiation for validation of cell subset-specific differential gene expression patterns with reservoir size (Fig. 1e, Supplementary data file 1).

### CD14+ monocytes have the most differentially expressed genes associated with reservoir size

PBMC from the 14 Thai male participants collected 48 weeks after ART initiation were assessed by scRNA-seq on the 10x Genomics platform using 5′ gene expression profiling. A total of 62,925 single cells passed quality filter and 19,581 genes were detected across all cell types from all participants. Cell clustering based on gene expression of lineage markers revealed 24 discrete populations (Fig. 2a, Supplementary Fig. 1). All major canonical immune cell populations in PBMC could be detected through gene expression, including cells from the innate, humoral, and cellular arms of the immune system (Supplementary Table 1). There were no significant differences in uniform manifold approximation and projection (UMAP) distributions or cell subset frequencies when comparing detectable versus undetectable reservoir groups (Supplementary Fig. 2a, b). Furthermore, no apparent differences in T cell receptor (TCR) or B cell receptor (BCR) clonal diversity or in BCR isotype distribution were observed between detectable and undetectable reservoir groups across all conventional T and B cell subsets captured in this analysis (Supplementary Fig. 3a–c). We performed differential expression analyses to identify genes whose expression showed quantitative differences between people with undetectable or detectable amounts of HIV DNA 48 weeks after ART initiation in all 14 participants. These analyses identified significant differences in gene expression between the two groups in 20 cell subsets. There were 224 unique significantly differentially expressed genes (DEG) which were independent of the size of the immune cell subsets. The cell types with the highest number of DEGs were CD14+ monocytes ($n = 78$), CD8 + $T_{CM}$ cells ($n = 51$), CD8 + $T_{EM}$ cells ($n = 46$) and CD16 + monocytes ($n = 38$) (Fig. 2b, Supplementary table 2). Analyses of the 224 DEG identified the top 20 significant pathways and processes collectively enriched across the different cell subsets (Supplementary Table 3). The top 3 significant gene ontology terms were lymphocyte activation, regulation of cytokine production, and cytokine-mediated signaling, with most of the pathway enrichments resulting from the DEG in monocytes. The only pathways that were significantly enriched in the detectable reservoir group were "lymphocyte activation" and "immune response-activating signal transduction" in CD4+ naïve T cells (Supplementary table 4). DEG with the greatest significance in different cell subsets are highlighted in Fig. 2c. The DEGs that were most significant, and with an average log fold change of >1, were thrombospondin-1 (*THBS1*) and interleukin-1 beta (*IL1B*) in CD14 + monocytes (Fig. 2c). The median expression of these two genes in CD14+ monocytes was significantly higher in the undetectable compared to the detectable reservoir group when assessed using a single-cell approach ($P_{adjusted} < 5e\text{-}324$ and $P_{adjusted} = 8.4e\text{-}197$ for *THBS1* and *IL1B*, respectively) or by participant-specific average gene expression analyses (rho = −0.7, $P = 0.005$ and rho = −0.87, $P = 5.49e\text{-}05$ for *THBS1* and *IL1B*, respectively) (Fig. 2d, e). These genes were consistently expressed at higher levels for individuals with undetectable reservoirs, whether measurements were determined by total HIV DNA in PBMC or only in CD4+ T cells (Supplementary Fig. 4a, b). CD14+ monocytes also had the most number of significant DEG, and the gene associations remained significant even when the outcome was HIV reservoir decay from week 0 (AHI) to week 48 (on ART) (Coefficient = 4.99e-04 $P_{adjusted} < 5e\text{-}324$ and Coefficient = 1.47e-04, $P_{adjusted} = 3.97e\text{-}104$ for *THBS1* and *IL1B*, respectively) (Supple-

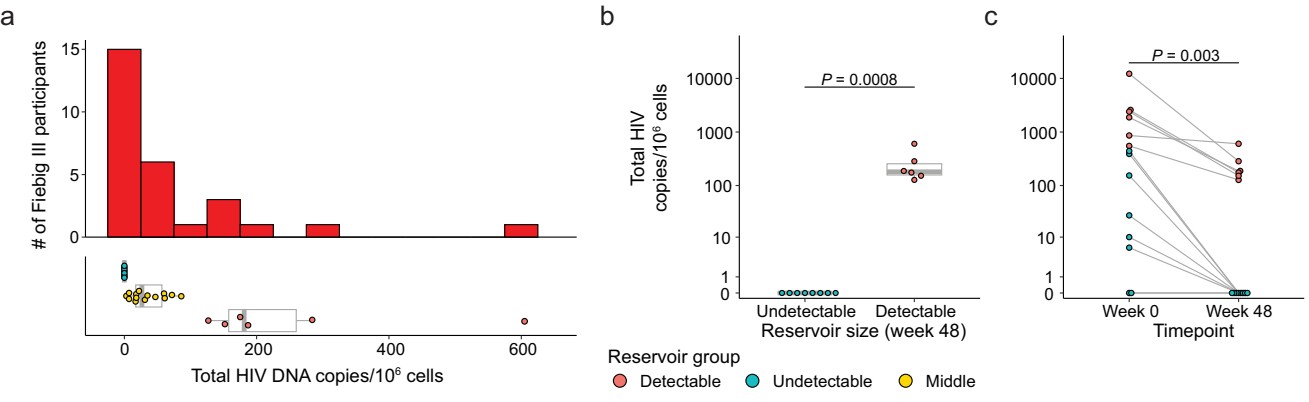

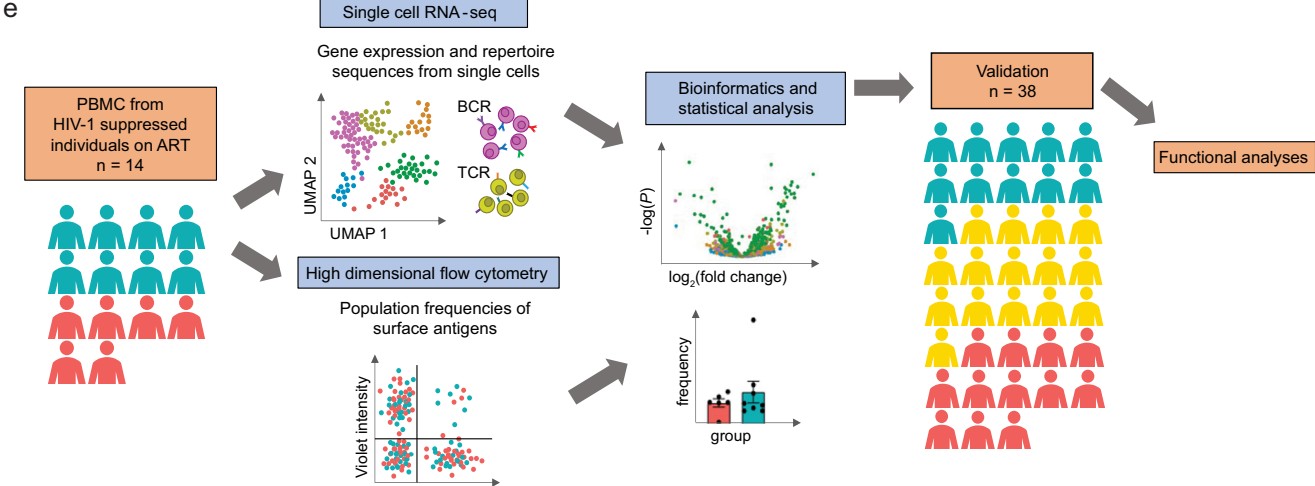

**Fig. 1 | Characteristics of study participants and experimental design.**
**a** Distribution of total HIV DNA in Fiebig stage III participants in RV254 at week 48 after ART initiation and their categorization into three groups based on reservoir size. The center of the box plot represents the median (50th percentile), while the upper and lower bounds of the box represent the lower and upper quartiles (25th and 75th percentiles, respectively). The whiskers extend from the box to the data points at most 1.5 times the interquartile range (IQR) from the lower and upper quartiles. **b** Selected participants from Fiebig stage III with extreme reservoir size phenotypes (undetectable = below LOD; and detectable = high) of cell-associated total HIV DNA in the RV254 Thai discovery cohort ($n = 14$). Significance was determined by the Mann-Whitney $U$ two-sided test. **c** Total HIV DNA decay between weeks 0 (AHI) and 48 (after ART initiation). Significance was determined by the Wilcoxon signed-rank two-sided test. **d** Phenotypes of participants comprising the detectable versus undetectable reservoir size categories. Mean values are shown for each group, NS not significant. Significance was determined by the Mann-Whitney $U$ two-sided test. **e** Single-cell RNA-seq and multiparameter flow cytometry were performed on all 14 participants. Additional validation by scRNA-seq was performed in an independent AHI cohort from the USA (A5354) ($n = 38$).

mentary Fig. 4c, d). A positive correlation between reservoir decay and participant-specific *THBS1* and *IL1B* expression was also observed (Supplementary Fig. 4e, f). Thus, from a screen of all peripheral blood cell populations, we observed the strongest correlations with reservoir size not for CD4+ T cell subsets, but for monocytes, which showed enrichment for activation pathways, and particularly increased expression of *THBS1* and *IL1B*, in the undetectable reservoir group.

## Monocyte-expressed genes in conjunction with central memory CD4+ T cell frequencies were associated with decreased reservoir size

To understand the association of monocyte gene expression with reservoir size, we used variation in cell frequency data obtained by multi-parameter flow cytometry to determine if specific populations varied between individuals. A total of 117 cell populations were

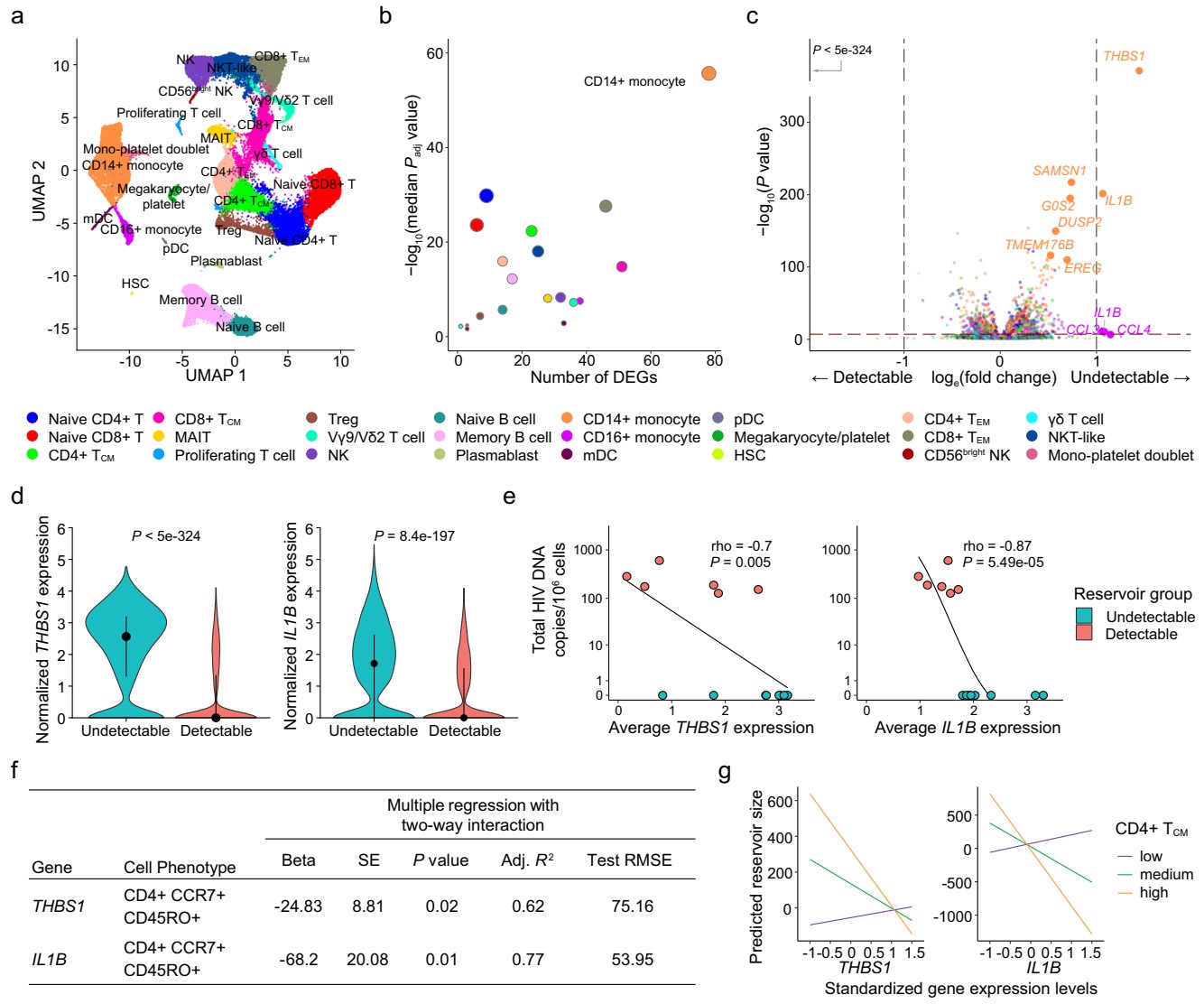

**Fig. 2 | Differentially expressed genes in monocytes associated with HIV reservoir size during ART. a** scRNA-seq identified 24 unique clusters of immune cell subsets. **b** CD14+ classical monocytes have the highest number of DEG between the detectable and undetectable reservoir groups. Significance was calculated by the Mann-Whitney $U$ two-sided test with Bonferroni correction. Circle color represents cell subset, while circle size indicates the corresponding number of cells. **c** Volcano plot shows DEG in all cell types. Genes with $P$ values that are significant after correction are indicated above the horizontal dotted line. Labeled genes have a $P < 10e-6$ and absolute average $\log_e$ fold change $\geq 1$ (vertical dotted lines) or $P < 10e-100$ and absolute average $\log_e$ fold change $\geq 0.5$. Significance was calculated by the Mann-Whitney $U$ two-sided test. **d** The most significant DEG in CD14+ monocytes comparing reservoir groups. Black dots represent the median normalized gene expression values ($\log_e$), and lines represent the interquartile ranges. Teal: undetectable reservoir, red: detectable reservoir. Significance was determined by the Mann-Whitney $U$ two-sided test with Bonferroni correction. **e** Participant-specific categorical analyses of the most significant DEGs. Normalized gene expression within CD14+ monocytes was averaged per participant and correlation was determined by the Spearman test ($n = 14$). The association direction is indicated via a monotonic trend line. **f, g** Table and interaction plots of multiple regression between standardized *THBS1* or *IL1B* expression in monocytes and predicted reservoir size with varying frequency of the CD4+ $T_{CM}$ population. Nominal $P$ values are indicated for the two-sided interaction analyses.

identified and annotated by cell surface marker expression including monocytes, DC, NK and B cell subsets as well as CD4+/CD8+ T cell activation, exhaustion and memory status from PBMC isolated at the same time as those used in scRNA-seq analyses (Supplementary data file 2). We first used a univariate linear regression analysis of the cell population frequencies and identified CD4+ T cells expressing CD39 on the cell surface as the only marker associated with significantly increased HIV DNA after adjusting for multiple testing (beta=16.9, SE = 3.6, $P < 0.001$, q = 0.08). These findings support the growing body of literature supporting a role for CD39 expression in HIV pathogenesis and latency[26–29]. To capture potential synergy between monocyte gene expression and specific cell populations, as defined by flow cytometry, that may affect the HIV reservoir, we performed a multivariate model with an interaction term. Measures of monocyte gene expression and

the frequency of immune populations were used to model the reservoir size and evaluate the two-way interaction of each of the 117 phenotypic population frequencies with the top two genes (*THBS1* and *IL1B* in CD14+ monocytes) that associated with decreased HIV reservoir size. There were several population-specific phenotypic markers whose frequencies increased in the presence of either *THBS1* or *IL1B* and were associated with lower reservoir size that were nominally significant (Supplementary Table 5). Of the 18 cell populations whose abundances correlated with either the expression of *THBS1* or *IL1B* and associated with decreasing reservoir size, two correlated with both of these genes (Supplementary Table 5). The two populations were subsets of central memory CD4+ T cells (CD4+ $T_{CM}$ cells; CD4+ CCR7+ CD45RO+) that were negative for PD-1 or HLA-DR surface markers (Supplementary Fig. 5a). Further grouping into other memory CD4+

phenotypes was not possible because of the absence of CD27 surface antibodies in the T cell flow cytometry panel. However, we observed no differences in frequencies of these CD4+ $T_{CM}$ cell subsets between participants with detectable or undetectable reservoir size (Supplementary Fig. 5b–d). Given the low frequencies of CD4+ $T_{CM}$ cells that are PD-1+ or HLA-DR+, we combined them with the frequencies of their respective negative populations and obtained their combined parent CD4+ $T_{CM}$ phenotype frequencies. A multiple regression model with two-way interaction also demonstrated a significant inverse relationship of CD4+ $T_{CM}$ frequency and monocyte *THBS1*/*IL1B* expression with reservoir size ($P = 0.02$ and $P = 0.01$ for *THBS1* and *IL1B*, respectively, Fig. 2f, g). In this interaction model the significance of *IL1B* over *THBS1* is further strengthened as shown by better accuracy (adjusted coefficient of determination (adj. $R^2$)) and deviation (test root mean square error (test RMSE)) metrics (Fig. 2f). Thus, increased expression of *THBS1* or *IL1B* in monocytes in the presence of higher frequencies of CD4+ $T_{CM}$ associated with decreased reservoir size. This suggests that changes in monocyte *THBS1* and *IL1B* expression affect the size of the reservoir via an indirect effect on CD4+ $T_{CM}$, which was the only cell type to show this interaction in PBMC populations measured by flow cytometry. Given that memory CD4+ T cells are preferentially infected by HIV, this was consistent with our single-cell RNA-seq findings from monocytes, indicating an effect on the latent reservoir.

### Association of *IL1B* expression in monocytes with smaller HIV reservoir size in an independent cohort using different measurements of HIV DNA

To verify the significance of our findings, we used an independent cohort of acutely treated PLWH from the USA (ACTG A5354). This cohort was comprised of 38 male participants of European and African ancestry, with treatment initiated during Fiebig stages III-V. Total HIV DNA reservoir was measured at week 48 after ART initiation. Variation in HIV DNA levels was observed within both the European and African population groups, and scRNA-seq was performed on samples from the week 48 timepoint (Fig. 3a, b). A total of 22 cell subsets were identified (Fig. 3c, Supplementary Table 1), the majority of which were consistent with the RV254 cohort from Thailand. In this cohort we expanded scRNA-seq analyses to all available participants with not only detectable or undetectable reservoir, but also the middle group by using HIV DNA measurements as a continuous variable in the MAST statistical framework[30]. CD14+ monocytes had the greatest number of DEG (%) associating with differences in reservoir size (Fig. 3d). In this single-cell MAST analysis *IL1B*, but not *THBS1*, was significant in CD14+ monocytes and validated the directionality seen in the RV254 cohort (Coefficient = -0.13, $P = 5.1e$-34). This was most likely because of the low number of cells expressing *THBS1* (5.6%) in this cohort, consisting of people of African and European ancestry, versus in RV254 (56%). In contrast, *IL1B* was still well expressed in the ACTG A5354 cohort (21%), though this was also lower than in RV254 (57%). A categorical analysis of all 38 participants also showed an inverse correlation of *IL1B* expression with reservoir size (rho = -0.42, $P = 0.009$) and a significant difference in *IL1B* between the extreme reservoir groups (undetectable n = 11, detectable n = 12) ($P = 0.006$) with a gradient in gene expression with all three groups included (Fig. 3e). In contrast the *THBS1* association with reservoir size was not significant (rho = -0.15, $P = 0.35$, categorical $P = 0.41$). Thus, regardless of viral subtype (B or CRF01_AE) or host background (Black, White, Thai), *IL1B* expression in monocytes had a significant inverse association with HIV reservoir size in both the discovery and replication cohorts. Further, IPDA® measurements of the persistent proviruses that comprise the reservoir were available in a subset of the ACTG A5354 study (n = 21), where intact or defective proviruses could be analyzed separately. Significantly higher frequencies of persistent intact proviruses compared to defective proviruses after 48 weeks of ART initiation were observed in this cohort of PLWH treated during AHI (Fig. 3f). Participant-specific analyses of *IL1B*

expression showed an inverse correlation with different forms of proviruses (Fig. 3g). Notably, when we harnessed the power of single cells using the MAST framework for continuous analyses, the inverse *IL1B* association was significant across most forms of the persistent provirus as measured by IPDA®, showing that findings were valid across total, intact, and defective proviruses (Fig. 3g).

### Transcriptional programs implicate NF-κB with differences in HIV-1 reservoir size

Given these significant effects of individual monocyte genes on reservoir size, we explored the broader consequences of transcriptional changes in monocytes using weighted gene co-expression network analyses (WGCNA) and identified nine modules of co-expressed genes within CD14+ monocytes from RV254 (Fig. 4a). The second largest module, M3, contained 452 genes including *IL1B*, and was significantly more highly expressed in the original Thai cohort from participants with an undetectable reservoir ($P_{adjusted} < 5e$-324, Fig. 4b). Comparing expression between the detectable and undetectable groups in the independent A5354 cohort using the M3 module genes identified in the Thai cohort, we confirmed that this module was similarly enriched in the cells from the undetectable group ($P_{adjusted} = 1.3e$-55, Fig. 4c). There were no other modules that were significantly associated with the reservoir size in both studies. Expression of the top 25 hub genes in this M3 module was generally higher in the undetectable than in the detectable reservoir group in both cohorts (Fig. 4d). The strength of this signature was further reinforced by the predicted interaction of the genes at the protein level (Fig. 4e). Gene ontology analyses showed that hub genes in the M3 module were enriched in several pathways, including inflammatory response, regulation of apoptosis, and NF-kappa B (NF-κB) signaling (Fig. 4f). "TNFα signaling via NF-κB" had the largest membership of hub genes from the M3 module. Although meaningful single-cell cell-cell interaction analysis to confirm the potential ligand-receptor interactions was not possible due to negligible expression of IL1B receptors in CD4+ T cells, we were able to investigate coordinated gene expression patterns in memory CD4+ T cells using WGCNA (Supplementary Fig. 6a). Complementing the findings in CD14+ monocytes, once again the same signaling pathway was also enriched in a module (M4) that was highly expressed in the undetectable reservoir group in the memory CD4+ T cell subset (RV254: $P_{adjusted} = 4.96e$-24, ACTG: $P_{adjusted} = 1.44e$-31) (Supplementary Fig. 6b–e), suggesting an effect on NF-κB signaling in the cell population which harbors the latent reservoir. *DUSP2, EIF1, NR4A2, PER1, SLC2A3, TNFAIP3* and *ZFP36* were enriched in this pathway in memory CD4+ T cells and had significantly higher expression ($P_{adjusted} = 3.36e$-12 to 4.84e-63) in the undetectable compared to the detectable group in the RV254 study. Except for *PER1*, all findings were similarly significant in the ACTG study ($P_{adjusted} = 8.37e$-03 to 5.42e-32) (Supplementary Fig. 6f). These pathways are consistent with the *IL1B* findings, suggest a broader change in the inflammatory homeostatic state, and may define a coordinated transcriptional change that accompanies *IL1B* expression differences which associate with reservoir size.

### IL1B activates NF-κB, enhancing productive HIV infection while inhibiting viral spread in vitro

Binding of IL1B to its IL1 receptor induces a signaling cascade ultimately leading to the activation of NF-κB[31]. This transcription factor plays a key role in LTR-mediated transcription of proviral DNA, and its stimulation is well known to reactivate latent HIV-1[32,33]. Thus, we explored whether activation of NF-κB in CD4+ T could explain why increased monocyte *IL1B* expression could reduce the size of the latent HIV reservoir. To assess whether IL1B activates NF-κB, we treated A549 NF-κB reporter cells, which express the secreted embryonic alkaline phosphatase (SEAP) reporter gene under the control of the IFN-β minimal promoter fused to five NF-κB binding sites, with IL1B,

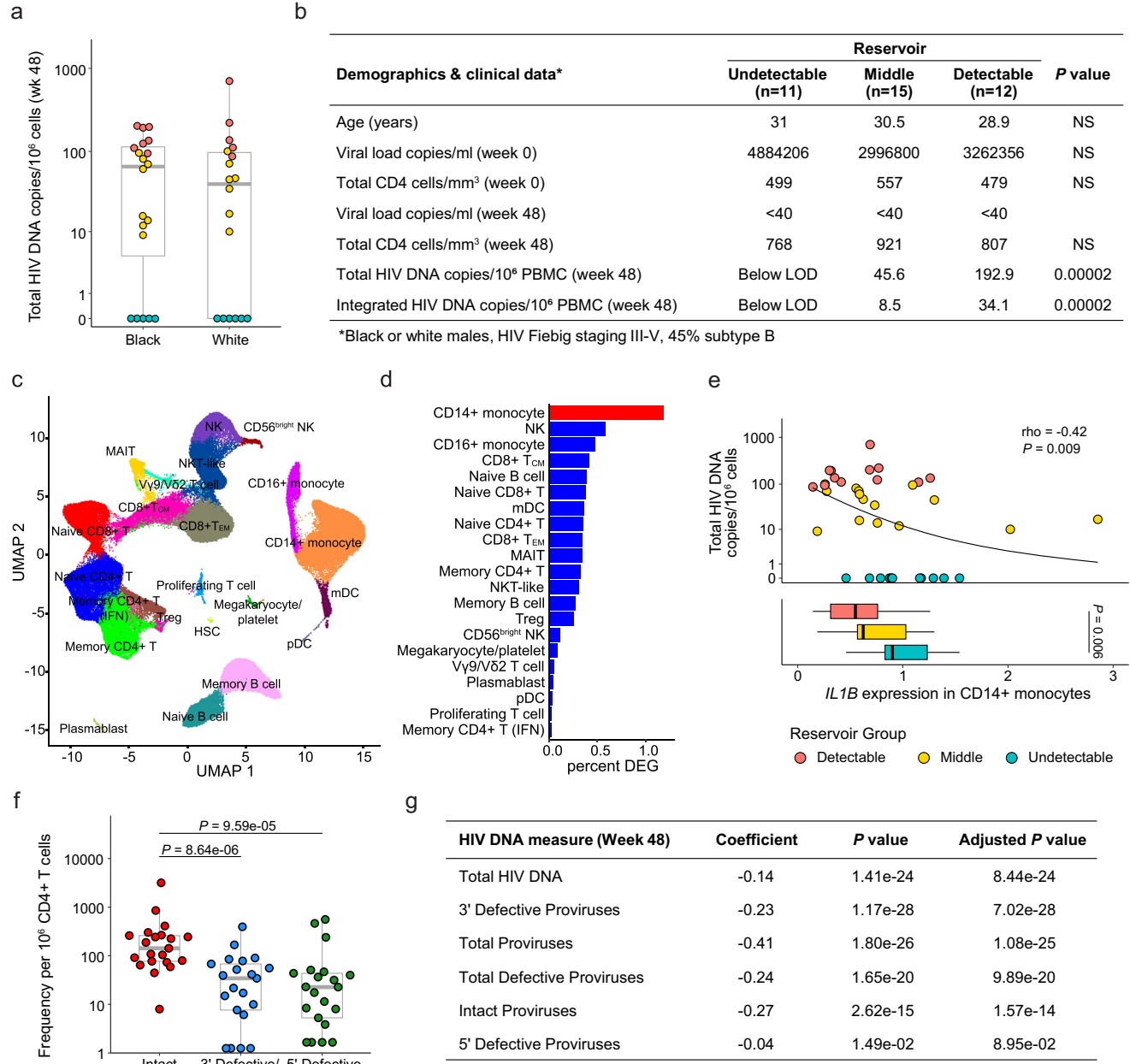

**Fig. 3 | Validation of *IL1B* association with smaller reservoir size from an independent cohort with a different infecting viral subtype across various Fiebig stages. a** HIV DNA levels vary within the A5354 subtype B cohort from the USA (n = 38). The participant samples used in this study are highlighted based on reservoir size. Red: detectable; teal: undetectable; and yellow: middle. Black and White indicate differences in the ancestry of the participants. **b** Characteristics of participants comprising the detectable, middle, and undetectable reservoir size categories (mean values) and the Mann-Whitney *U* two-sided test *P* values comparing the extreme phenotype groups are shown. HIV-1 subtype information was only available for a subset of the participants. NS: not significant **c** Dimensionality reduction plot of the different immune clusters in this cohort. **d** CD14+ monocytes have the highest number of normalized DEG associated with reservoir size using a continuous analysis including all 38 participants. **e** Participant-specific average *IL1B* expression in CD14+ monocytes categorized by total HIV DNA (n = 38). Spearman correlation *P* value and rho are shown. The directionality is indicated with a monotonic trend line. Box-whisker plots of the distributions of the three groups are shown below the x-axis. Significance was determined by the Mann-Whitney *U* two-sided test. **f** IPDA® measurements from a subset of the participants in this cohort (n = 21). Significance was determined by the Mann-Whitney *U* two-sided test. **g** *IL1B* association with different reservoir type measurements (rows) from the participants with IPDA measurements (n = 21). Significance was assessed using the MAST statistical framework (two-sided with adjustment for multiple corrections). For all box plots, the center of the box plot represents the median (50th percentile), while the upper and lower bounds of the box represent the lower and upper quartiles (25th and 75th percentiles, respectively). The whiskers extend from the box to the data points at most 1.5 times the interquartile range (IQR) from the lower and upper quartiles.

TNFα, LPS or medium only. We observed that IL1B increased NF-κB activity ~4-fold regardless of HIV-1 infection status (Fig. 5a). Next, we examined the ability of IL1B to activate NF-κB in primary human T cells. Degradation of inhibitory IκB proteins is a critical step in the activation of NF-κB, and their phosphorylation is one of the earliest and most

specific events in this process. Treatment with IL1B increased IκB phosphorylation and strongly reduced the overall levels of inhibitory IκB, indicating activation of NF-κB (Fig. 5b).

To more directly determine the impact of IL1B on latent and productive HIV-1 infection, we used the HIV molecular clone

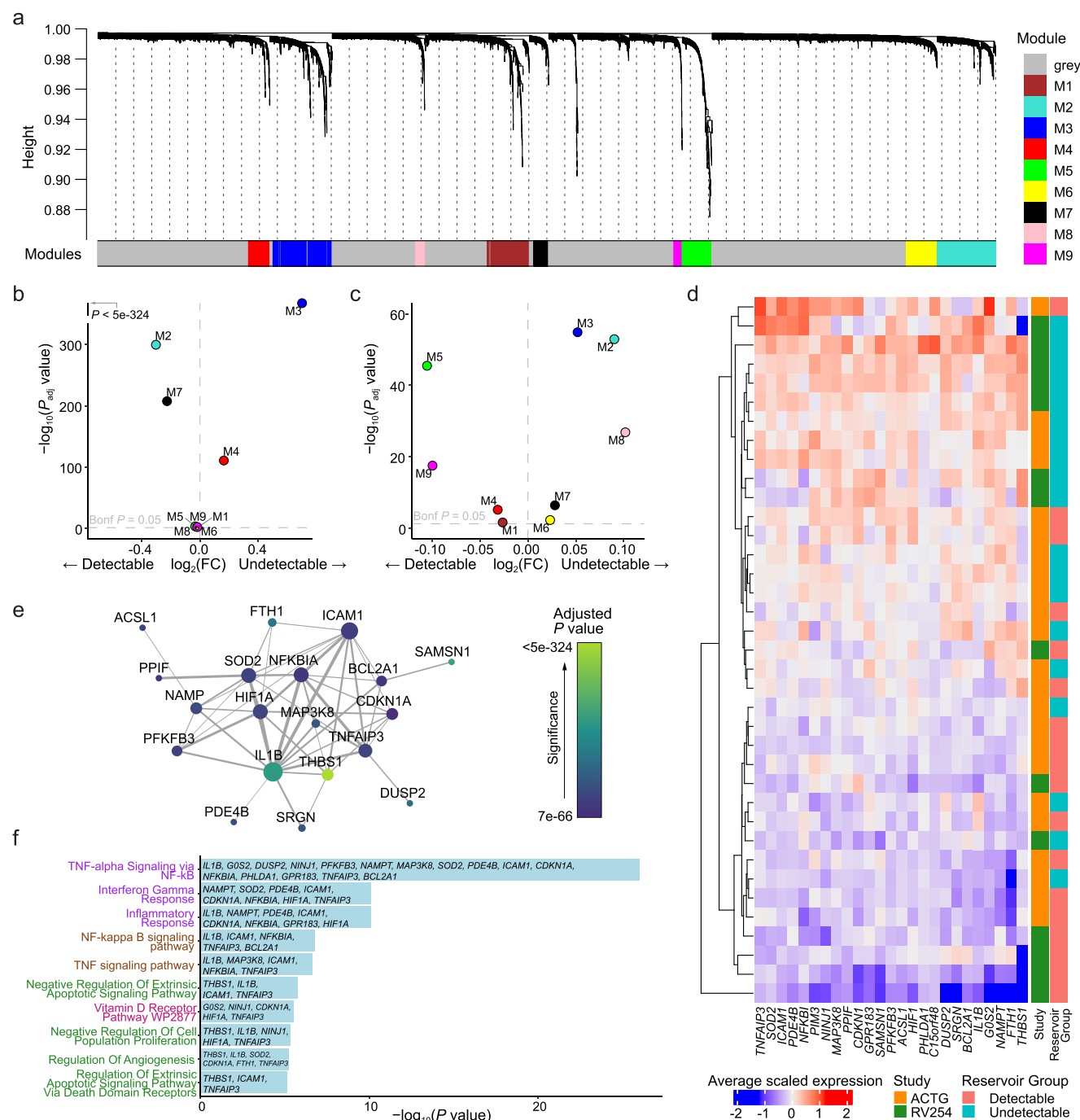

**Fig. 4 | Pathway analyses identified a distinct signature associating with reservoir size. a** Gene co-expression modules in CD14+ monocytes from the RV254 Thai study. **b** *IL1B* is in the M3 WGCNA module which was enriched in cells from RV254 participants with undetectable reservoir based on the top 25 hub genes in the module (detectable=6, undetectable=8). Significance was determined by the Mann-Whitney *U* two-sided test with Bonferroni correction. **c** Using the same module hub genes found in RV254, the M3 module was also enriched in cells from the undetectable reservoir participants in the A5354 cohort when HIV DNA levels were grouped categorically (detectable=12, undetectable=11). Significance was determined by the Mann-Whitney *U* two-sided test with Bonferroni correction. **d** Average expression of the 25 top hub genes from the M3 module was generally

higher in participants with undetectable reservoir in both cohorts. **e** Predicted protein interaction network of top 25 hub genes using the STRING protein data-base. Larger nodes have higher degree of connectivity; node color indicates sig-nificance in the categorical DEG comparison between the detectable and undetectable groups in RV254 CD14+ monocytes (*P* value determined using the Mann-Whitney *U* two-sided test with Bonferroni correction). **f** Gene ontology analyses of genes enriched in module M3 in CD14+ monocytes. Significance was determined using the Enrichr implementation of Fisher's exact test. *P* values were FDR corrected; the ten most significant genesets with adjusted *P* < 0.05 are shown. The color of y-axis labels indicates the originating database of the gene set: Hall-mark: purple, KEGG: brown, GO Biological Process: green, WikiPathways: pink.

pMorpheus-V5, which lacks a functional *env* gene but encodes all accessory proteins[34]. Cells productively infected with pMorpheus-V5 express V5-NGFR (Nerve Growth Factor Receptor) driven by the PGK (phosphoglycerate kinase) promoter, as well as HSA and mCherry

driven by HIV-LTR, while latently infected cells express only V5-NGFR. Activated PBMC from three healthy participants were treated with IL1B prior to, simultaneously, or after transduction with pMorpheus-V5 pseudo-typed with the Env proteins of WT X4-tropic NL4-3, an R5-

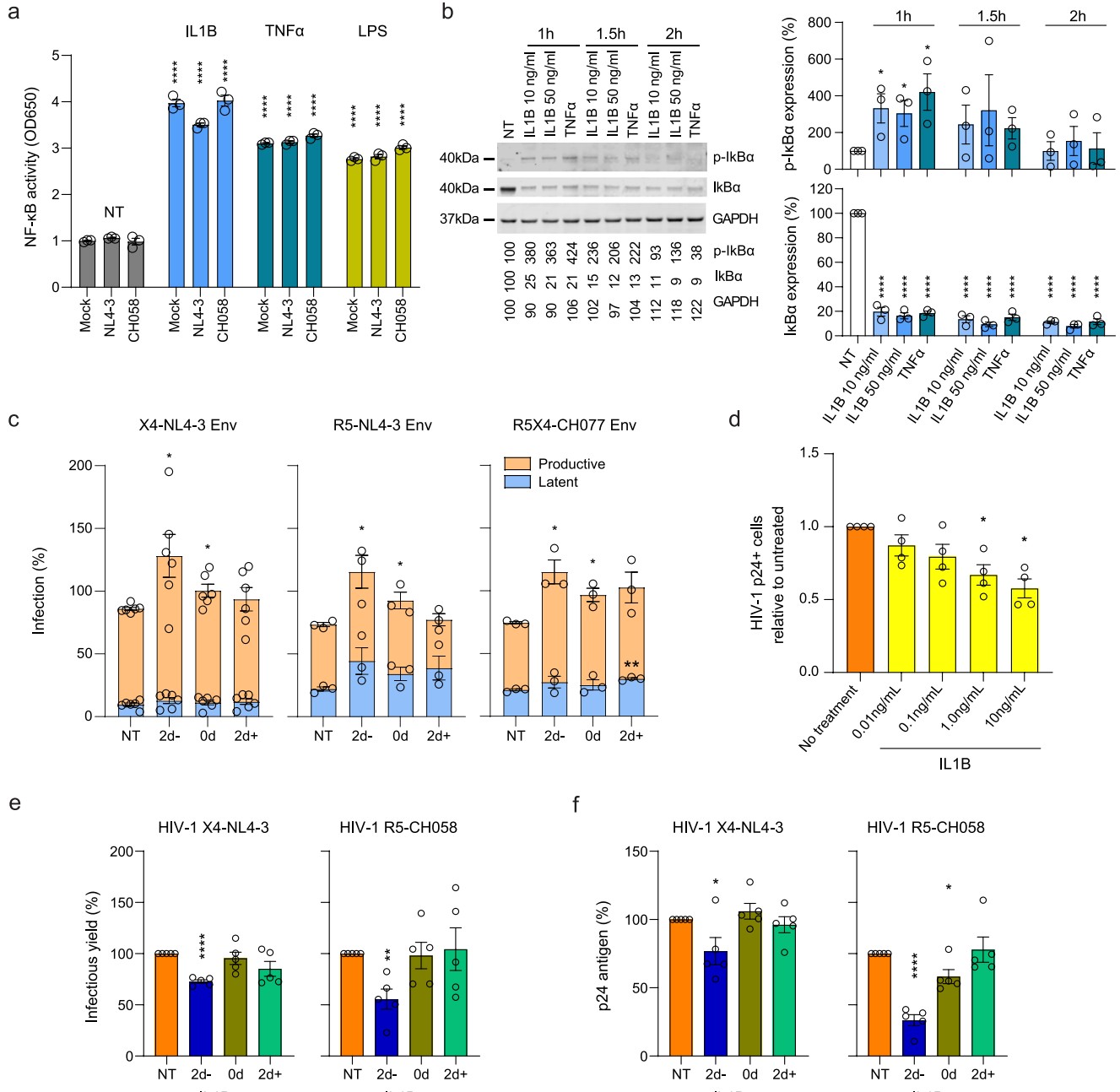

**Fig. 5 | In vitro IL1B activated NF-κB, increased HIV proviral transcription, and inhibited spreading infection. a** Effects of IL1B on NF-κB activity were assessed using A549 NF-κB reporter cells. Cultures were treated with IL1B, TNFα, or LPS and infected with VSV-pseudotyped NL4-3, CH058, or Mock control. After 24 hrs the Alkaline Phosphatase Blue Microwell assay was performed with OD650 values relative to the corresponding no treatment control (NT) reflecting NF-κB expression which is shown on the y-axis. The data presents an average of n = 3 independent experiments ±SEM. Blue: IL1B, teal: TNFα, yellow: LPS. **b** PBMC from n = 3 participants were treated with IL1B or TNFα and examined for IκBα phosphorylation as described in the methods section. Graphs present the protein expression from these n = 3 participants as mean ± SEM. Corresponding representative uncropped images are provided in the Source Data file. White: no treatment, light blue: IL1B 10 ng/ml, medium blue: IL1B 50 ng/ml, dark blue: TNFα. **c** Effects of IL1B in vitro when HIV was quantified after a single round of infection. Plots show the relative proportions of pMorpheus-V5 latently (blue) or productively (orange) infected PBMC in cultures treated with IL1B prior to, simultaneously, or after transduction with Env viral particles carrying the indicated Env protein. The background obtained from uninfected cells (Mock) was subtracted from the values

obtained for cells infected with the pMorpheus HIV-1 construct pseudo-typed with the indicated envelope proteins. The data represent n = 6 (X4) or n = 3 (R5 and R5X4) individual healthy participants, with error bars representing the average ±SEM. Corresponding gating strategies and representative plots are shown in Supplementary Fig. 7. **d** Effects of IL1B on spreading HIV-1 infection in cell culture. Using HIV-1 YU-2, bar plots display the relative p24-positive cell fractions after pre-treatment with increasing concentrations of IL1B (from 0.01-10.0 ng/ml, 10-fold increments) across n = 4 different participants. Significance was calculated using the Mann-Whitney *U* two-sided test. Orange: no treatment, yellow: different concentrations of IL1B treatment. **e, f** Bar plots display the average infectious virus yields (**e**) and p24 antigen levels (**f**) of n = 5 different participants at 4 days post-infection relative to the no IL1B treatment controls normalized to 100% ±SEM. Corresponding replication curves are shown in Supplementary Fig. 9. For all bar graphs shown, bar height and error bars represent mean values +/- SEM. Significant differences in all panels except (**d**), were determined using two-sided unpaired *t*-test analyses. Asterisks indicate statistical significance (**P* < 0.05, ***P* < 0.01, ****P* < 0.001, *****P* < 0.0001). Source data and exact *P* values are provided in the Source Data file.

tropic derivative thereof[35], and the dual R5X4-tropic CH077 transmitter-founder strain[36]. The frequencies of latently infected cells were generally lower compared to productive infection, and in most cases, not significantly altered by IL1B treatment. In contrast, the number of productively infected cells increased significantly ($P < 0.05$) for all 3 viruses when exposed to IL1B treatment during or 2 days prior to infection (Fig. 5c, Supplementary Fig. 7a–d).

NF-κB plays a complex role in HIV infection because it not only activates LTR transcription but also plays a key role in innate immunity and induces expression of numerous antiviral factors[37,38]. Indeed, pretreatment of stimulated PBMC with IL1B for 2 days resulted in a dose-dependent decrease in infection with the R5-tropic YU-2 virus as determined by the frequencies of sorted p24 + CD4+ T cells (Fig. 5d, Supplementary Fig. 8a). To further explore the effect of IL1B on spreading HIV-1 infection, stimulated PBMC from five participants were treated with IL1B prior to, at the same time, or after infection with HIV-1 NL4-3 or the transmitted founder CH058 molecular clone. Infectious virus production at 2, 4, 6, and 9 days post-infection was determined by p24 ELISA and infection of TZM-bl indicator cells. Infectious virus yields peaked at day 4 in most of the infected PBMC cultures (Supplementary Fig. 9). Two-day pretreatment with IL1B generally reduced viral replication compared to the untreated controls. Both infectious virus and p24 antigen production by NL4-3 and (more strongly) the primary CH058 strain were significantly ($P < 0.05$-0.001) reduced at 4 days post-infection (Fig. 5e, f, Supplementary Fig. 9). In comparison, only modest effects were observed when IL1B was added during or after infection, presumably because the induction of an antiviral state requires de novo synthesis of antiviral factors. Notably, the levels of cell death were low and did not differ significantly from the uninfected control (Supplementary Fig. 10).

IL1B is also known to affect the differentiation of CD4+ T cells into various subsets[39] that may differ in their susceptibility to HIV-1 infection. Notably, in the YU-2 infected cultures over 95% of p24+ cells expressed the CD45RO memory T cell marker, consistent with previous reports[40,41], and p24+ populations exhibited higher frequencies of both CD4+ T effector memory and transitional memory subsets (CD4+ T$_{EM}$ and CD4+ T$_{TM}$, respectively) compared to the p24- populations (Supplementary Fig. 8b, c). These findings underscore the importance of subset phenotypes for HIV infection and suggest that IL1B could alter the frequencies or phenotypes of HIV-susceptible CD4+ T cell subsets to modulate HIV reservoir size. We observed an IL1B dose-dependent increase in the frequency of CD4+ T$_{CM}$, accompanied by decreases in the frequencies of both CD4+ T$_{TM}$ and CD4+ T$_{EM}$, when PBMC from healthy participants were cultured in vitro (Supplementary Fig. 11a, b). Altogether, our in vitro data suggest that IL1B could decrease the HIV reservoir size in vivo through multiple mechanisms, including by promoting NF-κB-mediated activation of latent HIV, inducing innate antiviral factors, and changing the composition of T cell populations (Supplementary Fig. 12).

## Discussion

In this study, we used a single-cell approach to identify differences in host transcriptional profiles that associate with the size of the viral reservoir in acutely treated PLWH by screening extreme phenotypes of reservoir size. Recent bulk and single-cell transcriptomic studies have focused on the effects of differentially expressed host genes specifically in CD4+ T cells from PLWH on treatment[16,19,42], but it is important to also examine other cell populations that might influence the viral reservoir. We observed significant differences in the gene expression profiles of multiple immune cell subsets even after almost one year of complete viral suppression on ART, distinguishing participants with varying size of viral reservoirs. This finding was conserved across two cohorts comprising a total of 52 individuals and encompassing multiple host and viral genotypes. Significant differences were discovered in part due to accounting for potential confounders by selecting

participants matched for Fiebig stage at the time of ART initiation, pre-ART clinical parameters (viral load and CD4 counts), viral subtype, and sex prior to examining gene expression in single cells from participants in the Thai discovery cohort. These differences were generalizable to a distinct subtype B cohort comprised of participants with greater genetic variability and having IPDA® reservoir measurements. Frequencies of defective viruses in the ACTG study were lower than intact proviruses when measured by IPDA® which is not surprising considering the timing of sampling after ART initiation[43–47]. Regardless, the single-cell association of *IL1B* with reservoir size remained significant with different reservoir measures. These findings were also enabled by the use of scRNA-seq with its advantage, compared to bulk transcriptomics, that gene expression differences can be traced to specific cells rather than to "averaged" signals from heterogeneous populations.

We found that monocytes, specifically the CD14+ subset, showed the highest number of enriched pathways and DEGs, with *IL1B* being associated with differing reservoir sizes in two independent cohorts using various measurements of total, defective, and intact proviruses. IL1B is a potent proinflammatory cytokine[48], expressed in cells such as monocytes, neutrophils, B cells, and DCs, that is involved in a variety of cellular activities. Though *IL1B* was the most significant DEG present in both cohorts, we also detected a network of coexpressed genes that support a coordinated change in the monocyte transcriptional profile. Together, these findings are consistent with our previous observations that monocytes can play an important role both after vaccination and after treatment initiation when virus is suppressed[49]. In addition to assessing differences in host gene expression, frequencies of all major cell populations comprising PBMC were assessed by surface protein-based flow cytometry identifying CD39+ CD4+ T cells as associating with larger reservoir size, corroborating previous findings of various T cells expressing CD39 associating with poor HIV outcomes[26–29]. Since our focus was to evaluate frequencies of all 117 immune populations in the context of scRNA-seq gene expression to determine potential combined effects on peripheral blood reservoir size, we performed modeling using a multivariate two-way interaction analysis. We found from a total of 117 cell populations, that only frequency differences of a subset of central memory CD4+ T cells showed a significant interaction with monocyte gene expression and affected reservoir size. IL1B has been shown to act as a "licensing signal" for memory CD4+ T cells and has a role in the activation and function of memory CD4+ T cells[50]. It is possible that in this context central memory CD4+ T cells are potential targets of IL1B signaling, but more work is needed to establish if the interaction is specific to this subset rather than other CD4 cell types. Regardless, memory CD4+ T cells are known to harbor HIV and this interaction, consistent with our single-cell RNA-seq findings from monocytes, indicates an effect on the latent reservoir raising possible mechanisms which were explored in vitro. How exactly *IL1B* expression levels in monocytes influence the reservoir in vivo remains to be determined. We can speculate that the concept of "trained immunity", where epigenetic changes in monocytes or other innate cells by prior triggers such as infections and vaccines, could induce inflammation and immune memory as shown previously[51]. However, it is well established that IL1B mediates activation of NF-κB[31,52] and our in vitro data may explain a mechanistic link between increased *IL1B* expression in CD14+ monocytes and a reduced latent HIV reservoir size (Supplementary Fig. 12). The ILIB concentrations used in the in vitro assays are much higher compared to those found in plasma. It is well known, however, that local concentrations of IL1B are much higher than systemic concentrations across various tissues and pathological conditions, with systemic spillover usually being tightly regulated and minimal[53]. For example, average levels of 11.7 ng IL1B per ml have been reported in diseased periodontal tissues[54], which is similar to the concentration used in our in vitro assays (10 ng/ml). Besides IL1B, TNFα is one of the strongest endogenous inducers of NF-κB, and the pathway

"TNFα signaling via NF-κB" had the largest number of enriched genes in the M3 and M4 modules in CD14+ monocytes and memory CD4+ T cells, respectively. The key role of NF-κB in proviral HIV-1 gene expression has been known for decades. However, NF-κB also plays key roles in immunity and inflammation, inducing numerous antiviral factors[37,38]. Notably, NF-κB activates LTR transcription directly, while inhibitory effects require de novo synthesis of antiviral factors. Thus, HIV-1 and lentiviruses tightly regulate NF-κB activity to enable viral transcription while minimizing antiviral gene expression[55–57]. The induction of innate antiviral immunity by NF-κB may reduce viral reservoir seeding during acute infection. However, induction of proviral transcription by NF-κB is likely the more important mechanism in ART treated individuals, where viral replication is effectively suppressed, and induction of productive infection renders the latent reservoir susceptible to elimination. IL1B, TNFα and NF-κB all play complex roles in the survival, activation, and differentiation of T cells and other immune cells[39]. Thus, they may also impact the frequency of latently infected cells by more indirect mechanisms, such as shifts in the T cell subtype composition or cell survival. IL1B is best known for its role as a secreted cytokine. In some cases, however, it may also act in a cell-associated manner and the potential of IL1B-expressing CD14+ monocytes warrants further investigation. Notably, latency reversing agents (LRAs) that stimulate NF-κB have been extensively studied in shock-and-kill approaches and shown to reactivate HIV-1 from latency in both CD4+ T cell latency models and HIV-1-infected patient-derived cells[58–60]. Thus, it is tempting to speculate that, similar to TNFα, IL1B acts as a natural NF-κB inducing latency-reversing agent.

In addition to the strongest effect observed of higher *IL1B* levels, we also detected that *THBS1* in CD14+ monocytes associated with a smaller reservoir of infected cells. The association of *THBS1*, encoding for thrombospondin, with reservoir size was only observed in the RV254 Thai cohort, and not validated in A5354 participants from the USA, most likely attributable to differences in genetic ancestry, and so our mechanistic analyses focused on *IL1B*. However, population-specific associations are important and further studies are warranted to understand the effect of *THBS1* on reservoir size given that anti-HIV properties have been attributed to it previously[61]. Similarly these studies were limited to AHI cohorts, and exploration in the context of untreated chronic infection is also necessary. Another limitation to this single-cell approach is that it lacks equal power to detect effects across all cellular subsets and could miss effects in populations less frequent than monocytes. Additionally, how other monocyte genes or additional factors interact in vivo to influence the decreased reservoir size observed in participants with increased monocyte *IL1B* remains to be investigated further. Though intact proviral DNA measurements are still being adapted for use beyond subtype B[43] and therefore were not available for participants in the RV254 cohort, we were able to bridge our findings by leveraging IPDA® measurements of persistent intact and defective proviruses in the ACTG A5354 cohort. Although the strength of the associations differed, *IL1B* remained inversely associated with smaller reservoir size regardless of proviral intactness. It is plausible that different host factors exert effects depending on the form of the provirus and raises the need for additional in-depth investigations.

Direct clinical and translational implications of these findings remain to be investigated, but overall, our findings support that immune cells other than T cells can modulate the HIV reservoir in a clinical cohort, and that this effect may be influenced by specific genes and pathways. These findings were based on single-cell approaches and give rise to new hypothesis-driven questions that should be tested in other cohorts, where further confirmation of host cellular gene products and pathways involved in reservoir formation or maintenance may provide targets for therapeutic intervention and remission strategies. Our implementation of single-cell approaches enables broad screening for factors that impact reservoir size in ART-treated individuals in vivo. Altogether, our results suggest that IL1B-induced NF-κB activation is associated with undetectable viral reservoirs in vivo. Notably, several NF-κB activating LRAs have been shown to reactivate latent HIV in CD4+ T cells from PLWH[58]. However, no clinical trials of these LRAs have been conducted due to concerns about toxicity and risks associated with broad T-cell activation or inflammation. While adverse effects must be carefully considered, our results suggest that NF-κB-activating LRAs could serve as a potential latency-reversing strategy that may be effective in vivo.

## Methods
### Study design
Demographic and clinical data (viral load, CD4+ T cell counts, HIV DNA, HIV subtype, Fiebig stage) were available from 163 acutely-treated PLWH from the men who have sex with men (MSM) RV254/SEARCH010 cohort (NCT00796146) in Thailand[9,21,62]. For discovery analyses, scRNA-seq, immune receptor sequencing (TCR and BCR), and flow cytometry were performed on initially cryopreserved PBMC from 14 selected participants in this cohort and validated in an additional 38 male participants from the ACTG A5354 study (NCT02859558), a single-arm, open-label study to evaluate the impact of ART initiation during AHI conducted at 30 sites in the Americas, Africa, and Asia[63]. Samples from all participants from both prospective studies were collected at 48 weeks post-ART initiation. In a subset of participants, samples were also available from week 0 at AHI. Blood from healthy participants without HIV for in vitro experiments was obtained from the WRAIR 2567.05 institutional protocol. All participants from the aforementioned human studies provided informed consent and use of samples for research was approved by ethical review boards at the Walter Reed Army Institute of Research, USA, Chulalongkorn University Faculty of Medicine, Thailand and Advarra, USA.

### HIV Reservoir measurements
Total HIV DNA and integrated DNA were measured by quantitative PCR (qPCR)[10,25]. Briefly, pellets of PBMC and CD4+ T populations were suspended in 15 μl of Proteinase K lysis buffer per approximately 100,000 cells and digested for 18 h at 55 °C. Total HIV DNA was quantified using primers and a probe situated in the 5′-LTR, while primers and probe used for integrated DNA were situated in Alu and the 5′- LTR. ACH-2 cells (BEI Resources, NIAID, NIH; accession# ARP-349), which carry a single copy of the integrated HIV genome, were used to generate a standard curve for both assays. The cell input for each of the three replicates was approximately 100,000 per replicate (~300,000 total) and the lower limit of detection (LOD) of this assay was 3.3 copies/$10^6$ cells. LOD was calculated based on the number of cells analyzed followed by normalization to $10^6$ cells. Participants were grouped into detectable or undetectable reservoir based on the presence of total HIV DNA measured independently in both PBMC and CD4+ T cell populations, depending on sample availability for the latter. The presence of integrated HIV DNA, where available, was used as a confirmatory criterion for verifying the categorical groupings. The reservoir phenotype was defined as undetectable when both total and integrated HIV DNA were below the LOD. In contrast, the reservoir was defined as detectable when total HIV DNA > LOD.

### IPDA® assay
Accelevir Diagnostics performed HIV-1 intact proviral DNA assays (IPDA®) to discriminate between, and separately quantify, the frequencies of intact and defective persistent proviruses. The design and performance of this assay have been described previously[43,64]. Briefly, cryopreserved PBMC were thawed and CD4+ T cells were isolated and assessed for cell count, viability, and purity by flow cytometry. RNA-free genomic DNA was then isolated from the recovered CD4+ T cells, with concentration and quality determined by fluorometry and

ultraviolet-visible (UV/VIS) spectrophotometry, respectively. The IPDA® was performed, and data were reported as proviral frequencies per million input CD4+ T cells. These procedures were performed by blinded operators using standard operating procedures.

## Single-cell RNA library generation and sequencing

PBMC from the 14 RV254 participants on ART for 48 weeks were washed, resuspended in PBS plus 0.5% FBS, and simultaneously processed for scRNA-seq and flow cytometry. A total of 50,000 cells (at 1000 cells/μl) from each participant was set aside for scRNA-seq library construction and the remaining cells were used for flow staining as described later. The diluted PBMC suspensions were prepared for scRNA-seq using the Chromium Next GEM 5' Single Cell V(D)J Reagent Kit v1.1 (cat# 1000165) and the Chromium Controller (both 10x Genomics) per manufacturer's instructions. Briefly, targeting a recovery of 8000 cells/participant, samples were loaded into separate wells of Chromium chips. Amplified cDNA was used to make gene expression (GEX), TCR, and BCR libraries. The GEX library construction used a 14 or 16 cycle Sample Index PCR program, based on amplified cDNA concentrations. PBMC from the 38 A5354 participants were individually stained with TotalSeq-C anti-human hashtag antibodies (BioLegend), batched, and processed for gene expression (GEX) and hashtag oligo (HTO) libraries to improve cost-effectiveness[23]. Cells from each batch were loaded into 4 different wells of Chromium chips for targeted recovery of 16,000 cells/well.

Libraries from both studies were then assessed for quality and concentrations using the DNA High Sensitivity D5000 ScreenTape Assay (cat# 5067-5592) with the TapeStation (both Agilent), and subsequently pooled and quantitated with a MiSeq Nano Reagent Kit v2 (300 cycles) (Illumina; cat# MS-103-1001) sequencing run. Final libraries were sequenced using the NovaSeq 6000 S4 Reagent Kit (300 cycles) (cat# 20012866) on a NovaSeq 6000 instrument (both Illumina).

## Multiparameter flow cytometry

PBMC from 14 participants were stained with Aqua Live/Dead stain (cat# L34957), washed, and blocked using normal mouse IgG (cat# 10400 C) (both ThermoFisher). The cells from each participant were then split into four to run four different polychromatic flow panels using conjugated fluorescently labeled monoclonal antibodies against several surface markers to define B, T, Myeloid, and NK cell subsets (Supplementary data file 2, 3). For the T cells panel, cells were pre-stained with an MR-1 tetramer[65] prior to staining for additional surface markers. Following surface marker staining, cells were washed, permeabilized and fixed with eBiosciences FoxP3 Fixation/Permeabilization Set (ThermoFisher; cat# 00-5521-00). Cells were then washed, stained intracellularly, washed again, and analyzed using a FACS Symphony A5 (BD BioSciences). Data were analyzed with FlowJo v.9.9.6 or higher (Becton Dickinson).

## Virus production

HEK293T cells (ATCC; accession# CRL-3216) were transfected with 5 μg NL4-3 or CH058, or 5 μg pMorpheus plus 1 μg of designated envelope constructs using the TransIT-LT1 (Mirus; cat# MIR2306) transfection reagent per manufacturer's protocol. Infectious molecular clones of HIV CH058 and CH077 were kindly provided by Beatrice H. Hahn[56,66], while pMorpheus was provided by Viviana Simon[54]. Media were changed 24 hrs post transfection and virus stocks were collected 24 hrs later. PBMC were infected with freshly produced virus. The HIV YU-2 infectious molecular clone stock was obtained from the HIV Reagent Program (BEI resources; accession# ARP-1350).

## In vitro functional characterization

**Effects of IL1B on cell population frequencies and HIV infection.** PBMC were isolated from the blood of healthy participants by density centrifugation on a Ficoll-Paque Plus (GE Healthcare; cat# GE17144002) gradient and stimulated by anti-CD3/CD28 Dynabeads (Gibco; cat# 11132D) at a 1:1 ratio with the estimated CD4+ T cell population in PBMC (25% in total PBMC) in Complete Cell Culture Medium (RPMI 1640 Medium with GlutaMAX and HEPES (cat# 72400047), 10% fetal bovine serum (cat# A3840001), and penicillin/streptomycin (cat# 15140122)) (all Gibco) supplemented with 40 U/ml IL2 (cat# 130-097-743) and with or without recombinant IL1B (cat# 130-095-374) (both Miltenyi Biotec) at four different concentrations (0.01-10 ng/ml, at 10-fold intervals) for 4 days. Treated PBMC were either immediately analyzed using a FACSymphony A5 (BD BioSciences) to assess frequencies of T cell subpopulations, or infected with an R5 tropic molecular clone, YU-2, at a concentration of 1 μg of p24 per million cells and cultured for a further 2 days before assessing the relative frequencies of infected cells by flow cytometry (BD BioSciences).

**Effects of IL1B on HIV infectivity.** PBMC isolated from healthy participants were treated with 10 ng/ml IL1B concentrations at different times relative to HIV infection initiation: pre-treated 2 days prior to infection, added simultaneously, or added 2 days post-infection. Briefly, PBMC were isolated by Ficoll gradient centrifugation and cells were stimulated by PHA (ThermoFisher; cat# R30852801) in Complete Cell Culture Medium with 100 U/ml IL2 for 3 days. On day 1 post-isolation the required cells were set aside for IL1B treatment for 2 days prior to infections. On day 3 post-isolation, PBMC were infected by spinoculation (1200 x g, 2 hrs, 26 °C) with 150 ng of freshly produced NL4-3, or 500 ng of CH058 virus strains, per million cells. After spinoculation cells were washed 5 times with 1x PBS (Gibco; cat# 14190-094) and resuspended in fresh medium containing IL2 and IL1B per the schedule. Cells were cultured for an additional 9 days during which 400 μl of supernatant was removed every second day for determination of infectious virus yields. To determine infectious virus yield, 10,000 TZM-bl reporter cells (BEI Resources; accession# ARP-8129) per well were seeded in 96-well plates. The next day cells were infected in triplicate for 3 days with the collected supernatants. Three days post-infection the TZM-bl cells were lysed and *b-galactosidase* reporter gene expression was assessed with the Gal-Screen Reporter Assay System (Invitrogen; cat# T1028) per manufacturer's protocol using an Orion microplate luminometer (Berthold).

## Flow cytometry staining of pMorpheus infection

PBMC from healthy participants were isolated by using separation medium, stimulated for 3 days with PHA (2 μg/ml), and cultured in RPMI 1640 medium with 10% fetal calf serum and 10 ng/ml IL-2 prior to infection. Some of the cells were treated with IL1B 2 days before pMorpheus infection, some at the time of infection, and some 2 days after. Five days postinfection, cells were collected and surface-stained against V5 (V5 Alexa Fluor 647, ThermoFisher; cat# 451098). After 30 min incubation, cells were washed 3x with PBS, fixed with 4% PFA (ChemCruz; cat# sc-281692) for 30 min, and analyzed by flow cytometry (FACSCanto; BD BioSciences).

## Western blotting

PBMC were isolated as described and cultured for 3 days with IL2 and PHA. On day 3, cells were treated with IL1B (10 and 50 ng/ml) and TNFα (10 ng/ml) (Miltenyi Biotec; cat# 130-094-019) for 1, 1.5, and 2 hrs. To generate cell lysates, cells were washed in PBS and subsequently lysed in Western blot lysis buffer (150 mM NaCl (Sigma-Aldrich; cat# 1.06404.1000), 50 mM HEPES (Sigma-Aldrich; cat# 7365-45-9), 5 mM EDTA (Sigma-Aldrich; cat# E9884), 0.1% NP40 (USBio; cat# 19628), 500 μM Na3VO4 (Sigma-Aldrich; cat# 450243), 500 μM NaF (Sigma-Aldrich; cat# 201154), pH 7.5)[67]. After 5 min of incubation on ice, samples were centrifuged (4 °C, 20 min, 12,000 × g) to remove cell debris. Whole cell lysates were mixed with 4× Protein Sample Loading

Buffer (LI-COR, at a final dilution of 1×; cat# 928-40004) supplemented with 10% β-mercaptoethanol (Sigma-Aldrich; cat# M6250), heated at 95 °C for 5 min, separated on NuPAGE 4 ± 12% Bis-Tris Gels (Invitrogen; cat# 10472322) for 90 min at 100 V and blotted onto Immobilon-FL PVDF membranes (Merck Millipore; cat# IPFL00010). The transfer was performed at a constant voltage of 100 V for 1 h using a wet/tank transfer system (BioRad; cat#1703930). Proteins were stained with the following primary antibodies: phospho-IκBα (Cell Signaling; cat# 2859), IκBα (Santa Cruz; cat# sc-1643), and GAPDH (BioLegend; cat# 607902).

### NF-κB reporter assay

A549-Dual™ Cells (InvivoGen; cat# s549d-nfis) were seeded at a density of 20,000 cells per well on 96-well plates. Cells were treated on the following day with IL1B (10 ng/ml), TNFα (10 ng/ml), or LPS (1000 U/ml) (Invitrogen; cat#15536286) and infected or not with VSV-G pseudotyped HIV-1 NL4-3 or CH058 for 24 hrs, when the Quanti Blue assay was performed as described by the manufacturer (InvivoGen; cat# CRL-3216).

### Viability assay

Cells were harvested, washed once with PBS and stained for 15 min at room temperature in the dark with eBioscience fixable viability dye 780 (ThermoFisher; cat# 65-0865-14). Cells were then washed twice with PBS, fixed in 2% PFA for 30 min at 4 °C and analyzed by FACs.

### Bioinformatics analyses

**Sequence data processing.** Single-cell gene expression data from PBMC were generated using the 10x Genomics Cell Ranger pipeline (v3.0.0 - 3.1.0) (10x Genomics) per manufacturer's recommendations and the 10x Genomics human reference library (GRCh38 and Ensembl GTF v93). For the RV254 sequencing run without hashing, the average number of genes per cell was 1236 and the average number of unique molecular identifiers (UMI) was 3288. The mean read depth per cell was approximately 103,000-236,000 reads. The minimum fraction of reads mapped to the genome was 92.95% and sequencing saturation was on average greater than 94%. For the hashed A5354 sequencing runs, the average number of genes per cell was 1432 and the average number of UMIs for RNA transcripts was 4192. The mean read depth per cell was approximately 69,000-88,000 reads for the gene expression library and 9000-14,000 reads for the antibody library. The minimum fraction of gene expression reads mapped to the genome was 88.5% and RNA sequencing saturation was, on average greater than 89%. Downstream analysis of Cell Ranger outputs, including quality filtering, normalization, multi-sample integration, visualization, and DEG were performed using the R package Seurat (v3.1.1 - 4.3.0).

**RV254 gene expression processing.** Cells with mitochondrial percentages greater than 10% and cells that had <200 or >6000 expressed genes were removed from analyses. 62,925 cells remained after the quality control (QC) process. After log-normalization with a scale factor of 10,000, the top 2000 variable features within each sample were selected. We found integration anchors using dimensions 1:30 and integrated cells from all 14 participants. Shared Nearest Neighbor-based (SNN) clustering was performed using the top 30 principal components (PC) with a resolution of 0.5, and cells in the clusters were visualized by UMAP projection. Cluster marker genes were determined using Seurat FindAllMarkers and cluster identities were manually annotated using differentially expressed genes between the clusters and known lineage cell markers ( https://github.com/thomaslab-MHRP/scRNA-seq_annotation_resources/blob/main/GEX_markers_for_cell_annotation.md).

**A5354 demultiplexing and gene expression processing.** HTO expression matrices were normalized, demultiplexed, and assigned to specific participants using the methods described[23]. Negative cells and cells with greater than 10% mitochondrial gene expression were removed. Gene expression matrices (containing a total of 21,870 genes) for all 38 participants and for doublet cells were normalized. We performed reference-based integration using two participants from each ADT batch as references. Cells that were identified as doublets via hash demultiplexing and cells in clusters from an initial round of QC that were enriched for doublets or had high expression of *HBB* were removed, and SNN clustering at resolution 0.3 was performed on the remaining 140,172 cells. Clusters were visualized and annotated using lineage markers and differentially expressed genes similar to the process for RV254. No γδ T cell or monocyte-platelet aggregation clusters were identified, and memory CD4+ T cells were comprised of one large cluster and one smaller cluster with upregulation of interferon-induced genes, instead of subsets of CD4+ T cells as observed in RV254.

**Differential gene expression.** Categorical differential gene expression analyses within each cell type subset between the two reservoir groups was performed within Seurat using the Mann-Whitney *U* two-sided test with Bonferroni correction (n = 19,581). Genes that were not expressed in at least 10% of cells in either group or that did not have a log fold change of >|0.25| were excluded from consideration, as were mitochondrial and ribosomal protein genes. The MAST framework was implemented to examine correlation of gene expression of different cell subsets with the continuous log-transformed total HIV DNA measurements as the outcome[30]. Genes with expression frequencies <10% were removed before analyses. Results from each cell subset were corrected for multiple testing using the Bonferroni correction. Genes without a beta coefficient >|0.1| and additional manually curated genes were excluded from consideration. Continuous MAST analyses for a subset of 21 participants with IPDA® data was performed to see if *IL1B* remained significant using different reservoir measurement parameters. Participant-specific expression values were generated using Seurat's AverageExpression in CD14+ monocytes within participants on the log-normalized expression data.

**TCR/BCR sequence analyses.** TCR/BCR clonotype identification, alignment, and annotation were performed using the 10x Genomics Cell Ranger pipeline (v6.1.2; 10x Genomics) per manufacturer's recommendations. Clonotype alignment was performed against the Cell Ranger human V(D)J reference library 7.1.0 (GRCh38 and Ensembl GTF v94). The Cell Ranger clonotype assignments were used for both BCR and TCR Clonotype visualization and diversity assessments, and analyses were performed using R for IG chains within annotated B cell types (memory B cells, naïve B cells) or TRA/TRB chains within annotated T cell types (CD4+ or CD8+ $T_{CM}$, $T_{EM}$, and naïve T cells).

**Pathway analyses.** Further DEG lists characterizing the detectable and undetectable reservoir groups within cell subsets from RV254 were used to perform a multiple gene list analysis in Metascape to acquire the top 20 representative terms of the most significantly enriched pathways[68]. The genes comprising each of these 20 pathways were used as input lists to perform Gene Set Enrichment Analysis (GSEA)[69] when comparing the detectable and undetectable groups, along with an average expression matrix of all genes within each cell subset for each participant that was generated from the single-cell data. The GSEA results were filtered by normalized enrichment score (NES) ≥|1.4|, *P* < 0.001. For WGCNA-based pathway analyses, the CD14+ monocyte cell subset of the RV254 cohort Seurat object was used as input for coexpression analyses implemented in the single-cell R package, hdWGCNA[70,71]. Metacells and a signed network were constructed within participants using non-default parameters (k = 25, max_shared=10, and soft power=9). The top 25 hub genes for each of the resulting modules were used as a feature set for Seurat's

AddModuleScore to generate a score for each module within each cell. The Mann-Whitney *U* two-sided test was used to compare the expression of the module scores between cells in the detectable and undetectable reservoir categories. This module scoring and testing method for the same sets of genes from RV254 was applied to the CD14+ monocyte cell subset in the independent A5354 cohort. The average scaled expression of the 25 hub genes from the M3 module containing *IL1B* within both cohorts was used as input for the ComplexHeatmap tool[72]. Similarly, gene modules were identified in total memory CD4+ T cells and gene ontology analyses were performed using Enrichr[73]. The 25 hub genes for the M3 CD14+ monocyte module were used as input in a protein STRING DB pathway analysis[74]. The disconnected nodes were removed, and the resulting network was investigated for degree of connectedness and visualized in Cytoscape[75].

### Statistical analyses

The associations between 117 phenotypic flow cytometry population frequencies and reservoir size were assessed by univariate linear regression models and corrected for multiple testing using false discovery rate (FDR). Exploratory analyses including multiple regressions without adjusting for significance were also performed to evaluate the relationship between the reservoir size as the response variable and two explanatory variables: *THBS1/IL1B* and each flow cytometry cell population using machine learning methods in R[76]. Finally, multiple regression models were fitted with two-way interaction terms between *THBS1/IL1B* and each phenotypic population marker, to test whether the effect of *THBS1/IL1B* on decreased reservoir size differed depending on the frequencies of individual cell subsets. Interaction plots for *THBS1/IL1B* were made to illustrate how the relationship between *THBS1/IL1B* expression and reservoir size changes with different frequencies of combined CD4+ $T_{CM}$. The overall fitness of the simple regression models of the combined CD4+ $T_{CM}$ population was evaluated using the coefficient of determination, R-squared value ($R^2$), and Root Mean Squared Error (RMSE). For multiple linear regression of the CD4+ $T_{CM}$ cells, the goodness-of-fit was measured using both $R^2$ and adjusted $R^2$ along with RMSE. The prediction error of the combined CD4+ $T_{CM}$ cell models was estimated using Leave One Out Cross Validation (LOOCV) and the test RMSE value was reported. Assessment of model diagnostics carried out using both the gvlma() function in R and diagnostic plots (Q-Q plot for normality, residuals vs. fitted values for homoscedasticity, leverage plots for influential observations, variance inflation factors for multicollinearity; not shown) showed that the assumptions of the linear models were reasonable after removing one outlier identified using Cook's distance. All explanatory variables for all regression analyses were mean-centered, and plots show predicted measurements.

All comparisons were two-sided, using appropriate statistical tests for paired or unpaired analysis. Correlations were performed by Spearman's rank correlation coefficient and monotonic lines showed directionality. A two-sided *P* value of <0.05 was considered statistically significant for all statistical analyses. Bonferroni or FDR corrections were applied for multiple testing when appropriate. All descriptive and inferential statistical analyses were performed using R 3.4.1 GUI 1.70 build (7375) v3.0 and higher, and GraphPad Prism 8.0 statistical software packages (GraphPad Software, La Jolla CA).

### Reporting summary

Further information on research design is available in the Nature Portfolio Reporting Summary linked to this article.

## Data availability

The scRNA-seq gene expression datasets generated in this study have been deposited in the GEO database under accession code GSE220790 and GSE256089. Processed scRNA-seq data are available in Figshare

[https://doi.org/10.6084/m9.figshare.c.7074125]. Demographic and clinical data used in this study are provided in the Supplementary Data file 1. Source data are provided with this paper.

## Code availability

Code is available in Figshare at https://doi.org/10.6084/m9.figshare.c.7074125.

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

## Acknowledgements

We would like to thank Dr. Nicolas Chomont, University of Montreal, for supporting efforts to setup the HIV DNA assay at MHRP. We thank Gautam Kundu and Timothy Ezebuiro, MHRP, for bioinformatics and statistical support. Lilia Mei Bose and Hasset Tibebe, American University, assisted with functional data analyses. We acknowledge Joseph Puleo, Summer Zheng and Justin Ritz, CBAR, Boston, for providing IPDA® data. The MR1 tetramer technology was developed jointly by Dr. James McCluskey, Dr. Jamie Rossjohn, and Dr. David Fairlie, and the material was produced by the NIH Tetramer Core Facility as permitted to be distributed by the University of Melbourne. We thank the participants and staff of the RV254/SEARCH010 and ACTG A5354 cohorts. The views expressed are those of the authors and should not be construed to represent the positions or views of the U.S. Army or the U.S. Department of Defense (DOD), the U.S. Centers for Disease Control and Prevention, the U.S. Public Health Service, the National Institutes of Health, the Department of Health and Human Services, or the U.S. Government. This work was supported by a cooperative agreement (W81XWH-07-2-0067) between the Henry M. Jackson Foundation for the Advancement of Military Medicine, Inc., and the U.S. DOD. This research was also funded in part by the U.S. National Institute of Allergy and Infectious Disease (grants AAI20052001 to N.L.M; 5UM1AI126603-05 to S.V. and R15AI172610 to T.I). L.C.N. acknowledges funding from the NIH under grant UM1AI164565. The RV254/SEARCH 010 study is supported by cooperative agreements (W81XWH-18-2-0040) between the Henry M. Jackson Foundation for the Advancement of Military Medicine, Inc., and the U.S. DOD; and in part by the Division of AIDS (DAIDS), National Institute of Allergy and Infectious Diseases (NIAID), National Institute of Health (NIH) (grant AAI21058-001-01000). Antiretroviral medications for RV254 were donated by the Thai Government Pharmaceutical Organization (GPO), ViiV Healthcare, Gilead Sciences and Merck. The A5354 study is supported by the National Institute of Allergy and Infectious Diseases of the U.S. National Institutes of Health (UM1AI068636, UM1AI068618, UM1AI106701, and P30AI027757). F.K. is supported by the DFG (SFB 1279, SFB 1506, KI 548 21-1) and an ERC Advanced grant (Project 101054456). M.V. is also supported by the DFG (VO 2829 1-1). R.A. is supported by the Division of Intramural Research, NIAID, NIH. IPDA® testing is supported by the National Institute of Allergy and Infectious Diseases of the U.S. National Institutes of Health (grant U24AI143502). Antiretroviral medications for A5354 were donated by Gilead Sciences. Material has been reviewed by the Walter Reed Army Institute of Research and there is no objection to its presentation or publication. The investigators have adhered to the policies for protection of human participants as prescribed in AR 70–25.

## Author contributions

M.V., A.M., A.N., T.I., and F.K. contributed to functional assays. M.B., S.T., G.M.L., and J.C. contributed to HIV reservoir assays. L.K.Y., P.K.E., S.S., A.G., and R.T. contributed to omics and data analyses. M. Creegan., D.P-P., A.W., J.R.C., K.M., M.A.E., performed flow cytometry. C.S., N.P., N.L.M., M. Corley., L.C.N., B.S., S.J.K., J.A., E.S.D., J.W.M., M.L.R., R.A., S.V., T.A.C., provided clinical cohorts, data, and scientific strategy. F.K. and R.T. performed overall interpretation and study supervision. A.G., P.K.E., F.K., and R.T. contributed to the writing of the original manuscript, and all authors reviewed and edited the manuscript. R.T. conceptualized and led the overall project.

## Competing interests

The authors declare no competing interests.

## Additional information

Philip K. Ehrenberg [1,20], Aviva Geretz [1,2,20], Meta Volcic [3,20], Taisuke Izumi[1,2,4,5,6], Lauren K. Yum[1,2], Adam Waickman [7,17], Shida Shangguan[1,2], Dominic Paquin-Proulx [1,2], Matthew Creegan [1,2], Meera Bose[1,2], Kawthar Machmach[1,2], Aidan McGraw[4], Akshara Narahari [6], Jeffrey R. Currier [7], Carlo Sacdalan [8,9], Nittaya Phanuphak[10], Richard Apps[11], Michael Corley[12,18], Lishomwa C. Ndhlovu [12], Bonnie Slike [1,2], Shelly J. Krebs [1], Jintanat Anonworanich [1,2], Sodsai Tovanabutra[1,2], Merlin L. Robb [1,2], Michael A. Eller [1,2,19], Gregory M. Laird[13], Joshua Cyktor [14], Eric S. Daar[15], Trevor A. Crowell [1,2], John W. Mellors[14], Sandhya Vasan [1,2], Nelson L. Michael[16], Frank Kirchhoff [3] & Rasmi Thomas [1] ✉

[1]U.S. Military HIV Research Program, Center for Infectious Disease Research, Walter Reed Army Institute of Research, Silver Spring, Maryland, USA. [2]Henry M. Jackson Foundation for the Advancement of Military Medicine, Bethesda, MD, USA. [3]Institute of Molecular Virology, Ulm University Medical Center, Ulm, Germany. [4]Department of Biology, College of Arts and Sciences, American University, Washington D.C., USA. [5]District of Columbia Center for AIDS Research, Washington D.C., USA. [6]Department of Biology, College of Arts and Sciences, Saint Joseph's University, Philadelphia, PA, USA. [7]Viral Diseases Program, Walter Reed Army Institute of Research, Silver Spring, MD, USA. [8]SEARCH Research Foundation, Bangkok, Thailand. [9]Research Affairs, Faculty of Medicine, Chulalongkorn University, Bangkok, Thailand. [10]Institute of HIV Research and Innovation, Bangkok, Thailand. [11]NIH Center for Human Immunology, National Institutes of Health, Bethesda, MD, USA. [12]Department of Medicine, Division of Infectious Diseases, Weill Cornell Medicine, New York City, NY, USA. [13]Accelevir Diagnostics, Baltimore, MD, USA. [14]Department of Medicine, University of Pittsburgh, Pittsburgh, PA, USA. [15]Lundquist Institute at Harbor-UCLA Medical Center, Torrance, CA, USA. [16]Center for Infectious Disease Research, Walter Reed Army Institute of Research, Silver Spring, MD, USA. [17]Present address: Department of Microbiology and Immunology, State University of New York Upstate Medical University, Syracuse, NY, USA. [18]Present address: Department of Medicine, Division of Geriatrics, Gerontology and Palliative Care, The Sam and Rose Stein Institute for Research on Aging and Center for Healthy Aging, University of California, San Diego, CA, USA. [19]Present address: Vaccine Research Program, Division of AIDS (DAIDS), National Institute of Allergy and Infectious Diseases (NIAID), National Institutes of Health (NIH), Bethesda, MD, USA. [20]These authors contributed equally: Philip K. Ehrenberg, Aviva Geretz, Meta Volcic. ✉e-mail: rthomas@hivresearch.org

