## [Peer Review file · Nature Communications]

Single-cell analyses identify monocyte gene expression profiles that influence HIV-1 reservoir size in acutely treated cohorts

Corresponding Author: Dr Rasmi Thomas

Version 0:

Reviewer comments:

Reviewer #1

(Remarks to the Author)

Summary of findings:

The manuscript by Ehrenberg et al. identifies host gene expression patterns associated with the size of the HIV-1 reservoir at 48 weeks after ART initiation by using scRNAseq on PBMC from individuals who began antiretroviral therapy (ART) during acute infection. Separating study participants by HIV DNA reservoir size into subgroups with detectable vs. undetectable reservoirs, the genes whose expression distinguishes the two participant subgroups and the cell types in PBMC responsible for this differential expression are identified.

Among all cell subsets present in PBMC, monocytes are found to have the most differentially expressed genes when comparing individuals with detectable vs. undetectable reservoirs. THBS1 and IL1B expression in monocytes inversely correlate with reservoir size. Expression of THBS1 and IL1B in monocytes show different relationships with reservoir size in participant subgroups stratified by frequencies of central memory CD4+ T cell (Tcm) subsets in PBMC.

The relationship between IL1B expression and monocytes and several measures of total and intact HIV DNA reservoir size is then validated in a separate cohort with a different racial/ethnic composition. Details and potential functional explanations for these associations are explored using co-expression and other bioinformatic analyses.

Based on these analyses, in vitro experiments are undertaken to examine the association between monocyte IL1B and reservoir size mechanistically. In vitro data are presented confirming findings from previous reports showing activation of the NF- κ B pathway by IL1B. Effects of IL1B in in vitro HIV infection models are explored. The overall conclusion is that IL1B expressed by monocytes may drive reduction in reservoir size on ART by promoting latency reversal, which may lead to death of reservoir cells.

Overall assessment of quality and importance:

This is a strong study. The use of two different cohorts from well-conducted acute infection studies is appropriate and powerful, and experiments are well designed. The use of scRNAseq to localize differential gene expression to specific cell subsets in people with different reservoir sizes appears to have been highly effective, and the findings are interesting, make biological sense, and appropriately focus the cure field on how non-CD4+-T-cells may affect reservoir size.

Although the study is strong, this reviewer has significant concerns related to narrative, data analysis and interpretation, and other issues. These are detailed below.

Major concerns:

1. Would be cautious about concluding from the data presented here that monocyte expression of IL1B drives smaller reservoir size. The associations between scRNAseq data and reservoir size measures are convincing and likely important, but do not demonstrate causation. While in vitro experiments could help provide a mechanistic link, would note that some of the experiments here used 10 ng/ml of purified IL1B to activate NF- κ B. Is this a physiologic concentration of IL1B?
2. Consideration in the Discussion section of what might be the cause of higher monocyte expression of IL1B in people with undetectable reservoirs across racially and ethnically diverse cohorts would strengthen the manuscript.
3. It is interesting that CD8 Tcm and effector memory (Tem) cell subsets were second and third after CD14+ monocytes in numbers of DEGs between participant groups. Could this reflect a role for CD8-T-cell-mediated killing in controlling reservoir size?

4. Line 184 mentions CD4+ T cells expressing CD39 as the only association with reservoir size in a univariate analysis, but the manuscript does not mention CD39 again. Performing an initial univariate analysis and identifying one hit seems to warrant further attention. It seems a strange omission for the manuscript not to consider what this result might mean.
5. The link between IL1B in monocytes and CD4+ Tcm cell frequencies seems tenuous. Two specific concerns are noted. First, although regression lines in Fig2F show similar trends for associations of IL1B and THBS1 with reservoir size in participant subgroups classified by Tcm cell frequencies, each of these lines seems to have been generated from just a handful of datapoints. While showing regression lines to support claims is acceptable, the individual datapoints should also be shown. Second, the mechanism of proposed link between monocyte IL1B expression and Tcm cell frequencies is not well explored in the manuscript. How could it be that the association of monocyte IL1B and reservoir size works differently depending on total Tcm cell frequencies in PBMC? Referring to the Tcm subset as “previously implicated as harboring the latent HIV reservoir” is vague and also incorrect. All CD4+ T cell subsets appear to be capable of harboring HIV in latent form.
6. In Fig3E, some individuals with a medium-sized reservoir appear to have similar or even higher IL1B expression in CD14+ monocytes compared to those with undetectable reservoirs. This reader was expecting to see a gradient in IL1B expression, with undetectable > middle > detectable reservoirs. Additionally, when comparing Fig3E to 2E, the difference in IL1B expression between undetectable and detectable reservoirs seems more pronounced in the Thai cohort than in the ACTG cohort. When looking at individual data points in Fig3E, the levels of IL1B expression do not seem different. These issues should be addressed in revision.
7. Would it be possible to quantify IL1B in plasma from the study participants directly? If possible, and if remaining samples permit, then adding these data could strengthen the manuscript.
8. In Fig5C, treatment with IL1B two days prior to infection appears to increase productive infection. However, in Figs5D, E, and F, the same treatment seems to reduce p24 levels and overall infection. This apparent discrepancy should be addressed.

Other comments and questions:

1. Suggest streamlining the Introduction to focus immediately on the central question of the manuscript. This first version appears to jump around between topics of immunology and reservoir biology.
2. Avoid the use of general and undefined terminology like “unbiased,” “high-throughput,” “more sensitive,” and “broader scope and resolution.” These instances of “sales” language can be distracting for the reader, especially because scRNAseq has its own issues and certainly does not overcome all limitations of previous technologies. Recommend revising to express the rationale for use of scRNAseq more clearly and objectively.
3. In several figures, reservoir sizes in individuals with undetectable HIV reservoirs are plotted at zero per 10e6 cells. It would be more accurate to plot reservoir sizes in these individuals at the detection limits of the assays. In fact, the methods section states that, for HIV DNA quantification, “The cell input for each of the three replicates was approximately 100,000 per replicate (~300,000 total) and the lower limit of detection of this assay was 3.3 copies/10e6 cells.” This text appears to indicate that fewer than 10e6 cells were analyzed for each sample, and that the assay is not sensitive enough to detect a single copy. Apologies if this is misunderstood.
4. IL1B is a well-established marker of inflammasome activation. Was there any enrichment of inflammasome pathway in monocytes in individuals with undetectable reservoirs? Given this pathway’s involvement in HIV persistence, it would be interesting to know if IL1B induction is due to this pathway in individuals with undetectable reservoirs.
5. If both total HIV DNA and integrated HIV DNA were used in classifying participants, why are only total HIV DNA data shown in Fig1? Please consider showing the integrated HIV DNA measurements, as well.
6. Please clarify what is being plotted on violin plots in Fig2D. Are these all cells from all participants, or are these individual values at one per participant? Text and callout to Fig. 2D-E on lines 167-8 suggest the latter, but then p values in Fig2D panel appear to be those of the single-cell (i.e., the former) approach. If plotting single cells, the total number of cells plotted should be noted. If one-per-participant values, then scatter plots of the 14 points would be preferable to violin plots.
7. Please include color codes in the figure or legend for Fig1A,B,C.
8. Please provide a flow cytometry plot for latent and productive infection quantification for Fig5C.
9. Please detail the methods to include amount of plasmids used for in vitro infection.
10. What are the expression levels of NF-kB pathway genes in CD4+ T cell subsets from individuals with undetectable reservoir?
11. Line 162 states that “The DEGs that were most significant, with an average log fold change of >1. . . .” “Most significant” suggests magnitude of p value, but the sentence seems to intend to refer to fold change rather than p value. Would reword this.

Reviewer #2

(Remarks to the Author)

This study by Ehrenberg et al. is a small yet thoughtful investigation into host transcriptomic differences in acute/early-treated people with HIV (PWH) with “undetectable” versus “detectable” HIV reservoirs. To minimize potential confounders, the authors conducted their analysis on a highly homogeneous primary cohort of 14 virally suppressed male Thai participants who initiated ART during Fiebig Stage III of acute HIV infection, using PBMC samples collected at 48 weeks of ART. They further validated their findings in a secondary cohort of 38 virally suppressed early-treated male U.S. PWH, again focusing on PBMC samples collected at 48 weeks of ART.

The findings align with a recent cohort-based study of virally suppressed PWH treated during early and chronic infection, which demonstrated an inverse association between IL-1b, TNF-a, NF-KB, and HIV reservoir size in peripheral CD4+ T cells. This suggests that the downregulation of host proinflammatory responses in bystander cells (e.g., monocytes and uninfected CD4+ T cells) may underlie this inverse association (citation 42). Given these parallels, it remains unclear

whether the paper introduces a novel hypothesis or simply confirms these findings using single-cell sequencing data.

Major Comments:

1. What was the basis for the decision to dichotomize the outcome variable for this small sample size study (N=8 with "undetectable" vs. N=6 "detectable" HIV reservoir size by using total and integrated HIV DNA values)? This study design is problematic from a clinical and statistical standpoint. For example, for the former, the "undetectable" group might include wholly unique clinical phenotypes, such as elite controllers and/or post-treatment controllers (i.e., PWH able to control virus in the absence of therapy and PWH able to control virus after a period of time on ART, respectively). From a statistical perspective, these small comparator groups preclude regression modeling and inclusion of key clinical covariates that are predictive of HIV reservoir size (e.g., initial CD4+ T cell count, pre-ART viral load, etc.).
2. Because HIV total and/or integrated DNA (the primary endpoint/outcome) was not measurable in the (N=8) "undetectable" population, efforts to include these samples in any dose-response like analyses are also problematic. Specifically, the abundance of "tied undetectable values" from the lower end of the range may throw-off analyses such as linear or multiple regression (problems with leverage or weight) or even Spearman's correlation (ties complicate the calculation of exact p-values). The linear regression models should be replaced with (i) logistic regression models predicting if the outcome is detectable, (ii) a linear regression model conducted within the population with measurable HIV DNA reservoir values, or (iii) a more complicated model taking the censoring into account, like Tobit regression.
3. In Figure 2E, it appears that "average THBS1 expression" does not differ significantly between the two populations, whereas "average IL1B expression" may provide a more meaningful distinction. This observation suggests that THBS1, which does not replicate across cohorts, might be an artifact of the analysis method. Could the authors generate a similar figure to Figure 2E using IPDA data from the second cohort, where detection levels were not an issue?
4. For the primary cohort, HIV reservoir was measured as integrated HIV DNA and total HIV DNA. For the validation cohort, HIV reservoir was measured as intact HIV DNA and defective HIV DNA. The authors should explain potential biological vs. artifactual reasons for why these assay results had different findings in the acute Thai vs. the ACTG (early-treated) cohorts?
5. Could the authors please clarify the rationale for performing TCR/BCR sequencing? Given that HIV-specific assays were not conducted, it would be helpful to understand how global TCR and/or BCR differences are relevant to the study. This aspect was not clearly explained in the Methods or Results sections. If these findings were included solely as a feature of the 5' 10X Genomics kit, the authors might consider revisiting whether their inclusion is necessary.
6. Could the authors elaborate on the decision to include only acute and early-treated PWH in the study design? Additionally, how might the findings differ, if at all, in chronic-treated PWH?
7. Please clarify the statement in the Methods section, "Assessment of model diagnostics [...] showed that the assumptions of the linear models were reasonable after removing one outlier." How was the outlier identified? Which participant is represented by the outlier data? Besides improving the model fit, is there reason to believe this sample should be omitted? Did removing this sample affect the direction or significance of the results?
8. The introduction could benefit from a more comprehensive explanation of how the findings from the current study build upon or support the existing literature. While it is noted that these findings are primarily observed in monocytes—already recognized as the primary source of IL-1b in peripheral blood—this point alone may not sufficiently contextualize the study's contribution. It might be helpful to revise this section to incorporate key references, such as citation 42, and to explicitly describe how the current study advances or complements these prior studies.
9. The study is well-written and presents valuable findings; however, it could benefit from a more in-depth discussion of the clinical and translational implications of these results. Could the authors elaborate on their perspective regarding the observed inverse association between proinflammatory pathways and HIV reservoir size?

Minor Comments:

1. Overall, the directionality of the associations could be made clearer, as it is somewhat challenging to follow. Using more explicit terms such as "increase" or "decrease" (e.g., in host gene expression in relation to HIV reservoir) rather than "differential" or "change" would enhance clarity and help readers better understand the relationships being described, as they are not immediately intuitive.
2. A model or summary figure could be very helpful in illustrating the directionality of the observed associations. It would also enhance the understanding of the biological relevance of the results.
3. In Figure 2E (and 3E), the authors should remove the linear trend line, as it suggests a linear correlation. Since the figure appears to present a Spearman correlation, which does not necessarily imply a linear relationship, adjusting this could provide a more accurate representation.
4. It seems that there may be some missing column headings in Extended Table 1. Specifically, the labels for the columns "(copies/mL) (AHI)", "(cells/mm³) (AHI)", "(copies/mL) (ART)", and "(cells/mm³) (ART)" are unclear.

Reviewer #3

(Remarks to the Author)

In this manuscript, the authors illustrated the transcriptomic differences between PBMCs from patients with varying HIV DNA loads. They highlighted IL1B in CD14+ monocyte as a potential mediator of HIV reservoirs via the NF-kb signaling in CD4+ T cells. The manuscript is interesting and well-written. The analyses performed are rigorous, however require some methodological details. Additionally, to strengthen the findings further, I have some suggestions to fill the missing puzzles of the story.

Major comments:

1. Fig. S1 was far not adequate to support the cell type annotation in Fig. 2A and Fig. 3C. A full up-regulated gene list for each cell subset was required for both Fig. 2A and Fig. 3C. Additionally, the resolution of Fig. S1 is too low to match it with the cell identifications in Fig. 3C.
2. To support the authors' hypothesis on line 263 about the IL1B-IL1R signaling, it would be useful to perform a cell-cell interaction analysis to confirm the potential interaction from CD14+ monocyte: IL1B to CD4+ T cell: IL1R in the single-cell data.
3. It is necessary to show whether NF-kb activated differently in CD4 T cells from undetectable and detectable groups in scRNAseq data to support the hypothesis on lines 265-266.

Minor comments:

1. As shown in Fig. 1C, it seems that all patients of blue groups have lower HIV DNA copies than patients of red groups at week 0, therefore, I am a little surprised by the non-significant difference indicated in Fig. 1D.
2. Systematic comparisons/integration were suggested for the two scRNAseq datasets to show the consistence beyond only number of DEGs and IL1B. For example, a scatter plot showing the changes of the same genes in the two datasets to support the consistent alterations in CD14+ monocytes.

Reviewer #4

(Remarks to the Author)

Version 1:

Reviewer comments:

Reviewer #1

(Remarks to the Author)

Several concerns about the initial submission have now been addressed in the revision and rebuttal, but there are two major concerns about the revision.

1. The section of the Results entitled "Monocyte-expressed genes in conjunction with central memory CD4+ T cell frequencies were associated with decreased reservoir size" is very unclear.
 - a. Line 176 reports that there were 117 cell populations identified by flow cytometry. This sounds complicated. However, this reviewer does not see a list of the 117 cell populations (i.e., what are their markers?), nor is there any technical or conceptual explanation of the analysis process that defined them. Methods state only that "Data were analyzed with FlowJo v.9.9.6 or higher (Becton Dickinson)."
 - b. In response to the question from the initial submission about CD39, which was the only marker associated with HIV reservoir size in univariate analysis, the revision includes the following sentence: "These findings agree with earlier findings of the role of CD39 expression in HIV pathogenesis and latency (refs 26-28)." This reviewer was unaware of the role of CD39 expression in HIV pathogenesis and latency. Among the cited references, the first is a review article, the second is a brief phenotyping study providing little mechanistic insight, and the third is a more detailed study that is not unpacked at all here. As the only hit in a univariate analysis, the CD39 finding would seem to deserve a more thorough thought process.
 - c. Although the rebuttal letter and Methods provide some technical explanation of how the "multiple regression model with two-way interaction" analysis was performed, this reviewer remains very confused about why this type of analysis was selected, the logic of its design, and exactly what the findings are (measured or modeled). The Discussion should offer a detailed mechanistic explanation of the findings that will unify them for the reader, while acknowledging ways in which this explanation could be wrong. The Discussion section in the revision falls short of this standard.
2. Flow cytometry in Supplementary Fig. 7 and related results in Fig. 5 appear problematic. Supplementary Fig. 7 does not show data from mock-infected controls. Therefore, it is difficult to be sure how much of each small cell population is background signal and how much is real. Considering this, it is difficult to interpret small absolute changes in frequencies of

latently-infected and actively-infected cells in this model.

Additional minor comments are as follows.

1. This reviewer did not understand the meaning of Lines 164-166, which appear to report an association between THBS1 and IL1B expression in CD14+ monocytes and rate of reservoir decay. This reviewer does not see which portions of Supplementary Figs 4c and 4d refer to reservoir decay. This needs to be clarified.
2. In Fig 1c, the y-axis is not labeled. Also, the uppermost blue dot at Week 0 does not have a segment connecting to week 48, while the second uppermost blue dot appears to have two segments.
3. In Supplementary Fig. 4, please clarify the relationship between participants shown in panels a and b. Presumably, the 5 participants shown as detectable and 6 shown as undetectable in panel b were assigned to the same groups in panel a, i.e., reservoir detectability agreed between measurements made in CD4+ T cells and those made in PBMC. Assuming this is true, it would be clearer to combine all data in one graph. If not true, the discrepancy should be explained.
4. What is the meaning of the "-" character in Supplementary table 1, which reports frequencies of different cell populations in RV254 and ACTG cohorts?
5. Lines 219-220 state that while HIV DNA levels are correlated with IL1B expression in monocytes, they are not correlated with THBS1 expression in monocytes. Fig 3e should show the THBS1 data to support this statement.
6. If the authors consider it to be analytically appropriate, the manuscript might mention whether there were any genes that were differentially expressed between detectable and undetectable groups in the ACTG validation cohort but not the RV254 exploratory cohort.
7. Supplementary Fig 6a-6e are not mentioned in the manuscript text. All included display items should be called out in the text.

(Remarks on code availability)

Reviewer #2

(Remarks to the Author)

The authors have done a nice job of addressing my comments. I have no further comments.

(Remarks on code availability)

Reviewer #3

(Remarks to the Author)

My concerns have been acceptably addressed.

(Remarks on code availability)

The codes were well-documented and enough for reproducing the major findings.

Reviewer #4

(Remarks to the Author)

(Remarks on code availability)

Reviewer #5

(Remarks to the Author)

(Remarks on code availability)

Version 2:

Reviewer comments:

Reviewer #1

(Remarks to the Author)

The authors have addressed this reviewer's concerns. The following minor comments can be addressed editorially before publication.

Results section 3: Initially, the rationale for conducting the multivariate regression analysis was unclear, but the explanation provided has been helpful. Regarding the methodology/procedures, any further clarification for readers not expert in these statistical methods could be beneficial.

Results section 3: The authors now refer to a "subset of memory T cells" instead of "central memory T cell subsets." This reviewer would like to clarify that previous comments aimed to understand the rationale behind the experiment, and were based on the sense that a link to central memory was insufficiently explained. From this reviewer's perspective, it is appropriate to refer to central memory in the manuscript. Any further speculative explanations that might be offered to help readers imagine what the mechanism(s) of the link to central memory might be would strengthen the manuscript.

(Remarks on code availability)

REVIEWER COMMENTS

We thank all the reviewers for their comments. Our responses are shown in blue font and changes to the manuscript are highlighted in yellow in the text.

Reviewer #1 (Remarks to the Author):

Summary of findings:

The manuscript by Ehrenberg et al. identifies host gene expression patterns associated with the size of the HIV-1 reservoir at 48 weeks after ART initiation by using scRNAseq on PBMC from individuals who began antiretroviral therapy (ART) during acute infection. Separating study participants by HIV DNA reservoir size into subgroups with detectable vs. undetectable reservoirs, the genes whose expression distinguishes the two participant subgroups and the cell types in PBMC responsible for this differential expression are identified.

Among all cell subsets present in PBMC, monocytes are found to have the most differentially expressed genes when comparing individuals with detectable vs. undetectable reservoirs. THBS1 and IL1B expression in monocytes inversely correlate with reservoir size. Expression of THBS1 and IL1B in monocytes show different relationships with reservoir size in participant subgroups stratified by frequencies of central memory CD4+ T cell (Tcm) subsets in PBMC.

The relationship between IL1B expression and monocytes and several measures of total and intact HIV DNA reservoir size is then validated in a separate cohort with a different racial/ethnic composition. Details and potential functional explanations for these associations are explored using co-expression and other bioinformatic analyses.

Based on these analyses, in vitro experiments are undertaken to examine the association between monocyte IL1B and reservoir size mechanistically. In vitro data are presented confirming findings from previous reports showing activation of the NF- κ B pathway by IL1B. Effects of IL1B in in vitro HIV infection models are explored. The overall conclusion is that IL1B expressed by monocytes may drive reduction in reservoir size on ART by promoting latency reversal, which may lead to death of reservoir cells.

Overall assessment of quality and importance:

This is a strong study. The use of two different cohorts from well-conducted acute infection studies is appropriate and powerful, and experiments are well designed. The use of scRNAseq to localize differential gene expression to specific cell subsets in people with different reservoir sizes appears to have been highly effective, and the findings are interesting, make biological sense, and appropriately focus the cure field on how non-CD4+ T-cells may affect reservoir size. Although the study is strong, this reviewer has significant concerns related to narrative, data analysis and interpretation, and other issues. These are detailed below.

We thank the reviewer for the positive remarks and for highlighting its relevance in the CURE field.

Major concerns:

1. Would be cautious about concluding from the data presented here that monocyte expression of IL1B drives smaller reservoir size. The associations between scRNAseq data and reservoir size measures are convincing and likely important, but do not demonstrate causation. While in vitro

experiments could help provide a mechanistic link, would note that some of the experiments here used 10 ng/ml of purified IL1B to activate NF-kB. Is this a physiologic concentration of IL1B? Physiological relevance is a valid point. Importantly, systemic spillover of IL-1 β is usually tightly regulated and systemic concentrations may be undetectable despite high local concentrations at inflammatory sites across various tissues and pathological conditions. For example, levels of 11.7 \pm 2.9 ng/ml have been reported in diseased periodontal tissues. Thus, local concentrations of IL-1 β at sites of infection and inflammation approximate those used in our in vitro assays which agrees with the proposed role of IL-1 β in modulating reservoir size. But we also agree that the in vitro data only provides a mechanistic link to our primary findings and is now discussed explicitly in lines 368-376.

2. Consideration in the Discussion section of what might be the cause of higher monocyte expression of IL1B in people with undetectable reservoirs across racially and ethnically diverse cohorts would strengthen the manuscript.

To address this point, we now discuss that epigenetic changes in innate cells such as monocytes due to prior triggers such as infections and vaccines could induce inflammation and immune memory as described in this review (PMID: 32132681). This is now included in the discussion section (lines 365-368).

3. It is interesting that CD8 Tcm and effector memory (Tem) cell subsets were second and third after CD14+ monocytes in numbers of DEGs between participant groups. Could this reflect a role for CD8-T-cell-mediated killing in controlling reservoir size?

The authors agree that following CD14+ monocyte subset (17%), the second and third highest number of DEG associating with reservoir size are CD8 Tcm (11%) and CD8 Tem (9.9%) in the Thai study (Supp table S2). There is no doubt that other cell subsets including CD8 T cells are involved in responses to reservoir size, but we see that CD14+ monocytes are by far having the most contribution. This is further supported by the fact that CD14+ monocytes also have the greatest number of DEG even in the independent ACTG cohort (Fig. 3d). Hence our findings support that immune cells other than T cells can modulate the HIV reservoir in clinical cohorts.

4. Line 184 mentions CD4+ T cells expressing CD39 as the only association with reservoir size in a univariate analysis, but the manuscript does not mention CD39 again. Performing an initial univariate analysis and identifying one hit seems to warrant further attention. It seems a strange omission for the manuscript not to consider what this result might mean.

There is already a large body of literature supporting the role of CD39, an exhaustion marker, with increased HIV DNA levels, latency, immune activation and disease progression (PMID: 31681335, 35529864, 37910505). Our findings bolster this body of literature, and we now mention this in the results (line 180-181).

5. The link between IL1B in monocytes and CD4+ Tcm cell frequencies seems tenuous. Two specific concerns are noted. First, although regression lines in Fig2F show similar trends for associations of IL1B and THBS1 with reservoir size in participant subgroups classified by Tcm cell frequencies, each of these lines seems to have been generated from just a handful of datapoints. While showing regression lines to support claims is acceptable, the individual datapoints should also be shown.

In the discovery cohort, we aimed to boost the scRNA-seq analyses using multidimensional flow cytometry analyses and statistical methods. The insight we obtained from the interaction analyses provided a connection between blood CD14⁺ monocytes and a subset of the memory CD4 T cells which are the major reservoir. These results although from a small sample size, provided us impetus to validate the scRNA-seq analyses in additional samples from a replication cohort. The regression lines in Fig. 2f (now 2g) are not drawn directly from individual data points but are model-derived predictions based on regression analyses from all but one participant that fit the assumptions (n= 13). It is a plot of a model fit showing the predictions calculated using mean centered data to better interpret the interaction between IL1B/THBS1, and Tcm cell frequencies on reservoir size. We have now updated Figure 2f (now 2g) axis and legend to reflect the predicted reservoir values and standardized variables. Moving forward to ensure that interpretation is made cautiously, the revised draft explicitly states that the regression lines are based on model predictions to avoid misinterpretation in the figure legends and methods. We also reversed the order of Figures 2f and 2g to emphasize that the plot is based on the prediction values from the table.

Second, the mechanism of proposed link between monocyte IL1B expression and Tcm cell frequencies is not well explored in the manuscript. How could it be that the association of monocyte IL1B and reservoir size works differently depending on total Tcm cell frequencies in PBMC? Referring to the Tcm subset as “previously implicated as harboring the latent HIV reservoir” is vague and also incorrect. All CD4⁺ T cell subsets appear to be capable of harboring HIV in latent form.

We agree that all CD4 T cells subsets harbor latent HIV and have updated the introduction with appropriate references (lines 67-69). This exploratory regression analyses shows a link between CD14 monocyte gene expression and reservoir finding with frequency of a subset of memory CD4 T cells which is known to harbor HIV latent virus. Broadly, this link supports that gene expression differences in monocytes could influence reservoir size, as an interaction with cells harboring latent virus raises possible mechanisms. Although we address mechanisms in vitro, we agree how these may vary with CD4 T memory cell frequency in vivo remains to be established and now include this in the discussion (lines 363-366).

6. In Fig3E, some individuals with a medium-sized reservoir appear to have similar or even higher IL1B expression in CD14⁺ monocytes compared to those with undetectable reservoirs. This reader was expecting to see a gradient in IL1B expression, with undetectable > middle > detectable reservoirs. Additionally, when comparing Fig3E to 2E, the difference in IL1B expression between undetectable and detectable reservoirs seems more pronounced in the Thai cohort than in the ACTG cohort. When looking at individual data points in Fig3E, the levels of IL1B expression do not seem different. These issues should be addressed in revision.

The IL1B association is stronger in the Thai population, because samples were selected carefully to find the strongest associations with reservoir size by considering confounders such as time of HIV diagnosis (Fiebig stage) and ethnicity. This was not possible in the replication ACTG cohort from the US where samples from different ethnicity and variability in time of HIV diagnosis and ART initiation were present. As to the reviewer’s point, the largest difference in IL1B expression is observed when comparing the detectable versus undetectable groups, which we call the reservoir extremes in the paper (lines 81, 116, 225, 328) without the middle group. A gradient is observed in the ACTG study with significant differences between the detectable and

undetectable groups at the participant level analysis (P=0.006). Fig. 3e is now revised to clarify this further (lines 220-227).

7. Would it be possible to quantify IL1B in plasma from the study participants directly? If possible, and if remaining samples permit, then adding these data could strengthen the manuscript.

We have tried to measure secreted protein levels using three assays of ELISA, Luminex and SomaScan. No significant differences between the IL1B levels comparing AHI versus ART, or when samples were stratified by HIV DNA levels were observed.

Response figure: IL1B measured in plasma using 3 different methods Top left panel: ELISA and Luminex. Top right panel: SomaScan. Bottom panels: IL1B plasma measured by Luminex (left) and SomaScan (right) when stratified by HIV DNA levels.

However, none of the assay results (IL1B levels) from the plasma samples correlated with each other even when analyzed within the same timepoint, pointing to the unreliability of these assays. Also, IL1B acts locally and it is possible serum levels of IL1B may not correlate well with IL1B concentrations which are generally higher in tissues where it is produced and where the bulk of HIV infected cells reside. Therefore, we feel that our in vitro analyses of the impact of soluble IL1B on HIV are still useful in providing a plausible explanation for how IL1B expressed by monocytes might affect viral reservoirs as seen in our in vivo data. In our in vitro studies, we focused on the cell-free form, since this cytokine is typically secreted and best known for its role as soluble factor. Overall, we think that these IL1b measurements in plasma are unreliable and detract if included in the revision because systemic concentrations are irrelevant for function (lines 371-376).

8. In Fig5C, treatment with IL1B two days prior to infection appears to increase productive

infection. However, in Figs 5D, E, and F, the same treatment seems to reduce p24 levels and overall infection. This apparent discrepancy should be addressed.

We think that this is only a seeming difference and explained by differential timing of opposing IL-1 β induced NF-kB dependent effects. NF-kB is rapidly induced and directly activates HIV-1 LTR transcription. Thus, IL-1 β treatment has enhancing effects in short-term single round HIV infection assays, like the one shown in Figure 5c (because of direct, fast NF-kB-mediated activation of HIV-1 transcription). However, NF-kB also plays a key role in the antiviral immune response and induces the expression of numerous innate immunity factors. Setting cells in an antiviral state requires more time than HIV-1 LTR activation because it requires *de novo* synthesis of antiviral factors. Thus, IL-1 β induced NF-kB is expected to inhibit spreading HIV infection (as those shown in Figures 5d-f) because it induces an antiviral state in uninfected bystander cells. Notably, both these effects are expected to reduce reservoir size because while IL-1 β initially enhances HIV-1 transcription through rapid NF-kB activation, it also eventually induces an antiviral state in neighboring cells, limiting the viral spread. This is discussed in the revised manuscript (lines 380-388) and illustrated in our model in Figure S12.

Other comments and questions:

1. Suggest streamlining the Introduction to focus immediately on the central question of the manuscript. This first version appears to jump around between topics of immunology and reservoir biology.

Given the nascent nature of scRNA-seq and for the manuscript to be relevant to a wide audience, we provided broader introduction covering biology and immunology of the reservoir. We have streamlined to avoid the impression of jumping around.

2. Avoid the use of general and undefined terminology like “unbiased,” “high-throughput,” “more sensitive,” and “broader scope and resolution.” These instances of “sales” language can be distracting for the reader, especially because scRNAseq has its own issues and certainly does not overcome all limitations of previous technologies. Recommend revising to express the rationale for use of scRNAseq more clearly and objectively.

We did not mean for certain terms to be distracting, but used these adjectives to merely explain why scRNA-seq is advantageous over previous methods and the rationale for using this method over previous assays for specific questions such as investigating associations with a phenotype. Single-cell RNA-seq is unbiased as it is able to assess transcriptome-wide gene expression in all cell types present in the sample and avoids focusing on specific cell types based on prior knowledge as we ourselves have done previously due to limitations of bulk RNA-seq and cited here (Ehrenberg et al. 2019). The 10x Genomics technology is reasonably high throughput as >1000s of cells can be assessed simultaneously for gene expression and thus provides broader scope and resolution over other methods that we have used for immune-profiling including bulk RNA-seq and flow cytometry. We have however removed most mention of these terms in the paper.

3. In several figures, reservoir sizes in individuals with undetectable HIV reservoirs are plotted at zero per 10e6 cells. It would more accurate to plot reservoir sizes in these individuals at the detection limits of the assays. In fact, the methods section states that, for HIV DNA quantification, “The cell input for each of the three replicates was approximately 100,000 per replicate (~300,000 total) and the lower limit of detection of this assay was 3.3 copies/10e6

cells.” This text appears to indicate that fewer than 10e6 cells were analyzed for each sample, and that the assay is not sensitive enough to detect a single copy. Apologies if this is misunderstood.

This assay is based on a qPCR method developed in Dr. Nicolas Chomont’s lab (PMID: 25122785). The limit of detection (LOD) was calculated based on the number of cells analyzed followed by normalization to 10e6 cells, which we now include in the methods for clarification (lines 451-452). As established previously by authors using this method, the ‘undetectable’ was defined as 0 copies/10e6 cells and plotted accordingly in the following examples (PMID: 27637172 (Fig. 3), 37708852 (Fig. 3), 23718762 (Fig. 2), 36465028 (Fig. 1)).

4. IL1B is a well-established marker of inflammasome activation. Was there any enrichment of inflammasome pathway in monocytes in individuals with undetectable reservoirs? Given this pathway’s involvement in HIV persistence, it would be interesting to know if IL1B induction is due to this pathway in individuals with undetectable reservoirs.

Although we did not directly find inflammasome pathways as top hits in our analyses of genes in the undetectable group (Fig. 4f), per the reviewer’s suggestion we looked for inflammasome-related pathways by searching for “inflammasome OR inflammasomes” terms in the MSigDB database containing Canonical Pathways (BioCarta, KEGG Legacy, KEGG MEDICUS, PID, Reactome and WikiPathways), Gene Ontology sets (GO) and Hallmark.

Term ID	P _{adj}	-log ₁₀ (P _{adj})		IL1B	TNFAIP3	NFKB1	CXCL8	CXCL2	NLRP3
		0	≤16						
KEGG_NOD LIKE RECEPTOR SIGNALING PATHWAY	4.002 × 10 ⁻²	█		█	█	█	█	█	█
KEGG_MEDICUS_PATHOGEN_SHIGELLA_IPAH7.8_TO_NLRP3_INFLAMMASOME_SIGNALING_PATHWAY	4.998 × 10 ⁻²	█		█	█	█	█	█	█
KEGG_MEDICUS_REFERENCE_NLRP3_INFLAMMASOME_SIGNALING_PATHWAY	7.329 × 10 ⁻²	█		█	█	█	█	█	█
KEGG_MEDICUS_REFERENCE_NON_CANONICAL_INFLAMMASOME_SIGNALING_PATHWAY	1.003 × 10 ⁻¹	█		█	█	█	█	█	█
WP_ACTIVATION_OF_NLRP3_INFLAMMASOME_BY_SARSCOV2	1.003 × 10 ⁻¹	█		█	█	█	█	█	█
KEGG_MEDICUS_REFERENCE_NALP12_INFLAMMASOME_SIGNALING_PATHWAY	5.954 × 10 ⁻¹	█		█	█	█	█	█	█
KEGG_MEDICUS_REFERENCE_NLRP4_INFLAMMASOME_SIGNALING_PATHWAY	5.954 × 10 ⁻¹	█		█	█	█	█	█	█
KEGG_MEDICUS_REFERENCE_NLRP1_INFLAMMASOME_SIGNALING_PATHWAY	5.954 × 10 ⁻¹	█		█	█	█	█	█	█
KEGG_MEDICUS_REFERENCE_PYRIN_INFLAMMASOME_SIGNALING_PATHWAY	5.954 × 10 ⁻¹	█		█	█	█	█	█	█
KEGG_MEDICUS_REFERENCE_REGULATION_OF_NLRP3_INFLAMMASOME_SIGNALING_PATHWAY_NLRP3_INHIBITION	5.954 × 10 ⁻¹	█		█	█	█	█	█	█
WP_NANOMATERIALINDUCED_INFLAMMASOME_ACTIVATION	7.997 × 10 ⁻¹	█		█	█	█	█	█	█
GOCC_NLRP3_INFLAMMASOME_COMPLEX	8.953 × 10 ⁻¹	█		█	█	█	█	█	█
GOBP_INFLAMMASOME_MEDIATED_SIGNALING_PATHWAY	1.000	█		█	█	█	█	█	█
REACTOME_INFLAMMASOMES	1.000	█		█	█	█	█	█	█
REACTOME_THE_NLRP3_INFLAMMASOME	1.000	█		█	█	█	█	█	█

Response figure: Enrichment of inflammasome related pathways in the “undetectable” DEGs is not as strong as the enrichment of TNF-alpha signaling via NF-kB pathway that we saw in the M3 hub genes.

This identified 17 gene sets that were interrogated for differences with phenotype. Genes that were higher in undetectable group than detectable in CD14 monocytes (P_{adj} < 0.05, logFC ≥ 0.25, and expressed in at least 10% of cells in either group) were used to test for enrichment. The KEGG NOD-like receptor signaling pathway was significant but did not have high gene membership. This indicates that this pathway does not have a strong association with the undetectable phenotype compared to the unsupervised analyses we performed in Figure 4.

5. If both total HIV DNA and integrated HIV DNA were used in classifying participants, why are only total HIV DNA data shown in Fig1? Please consider showing the integrated HIV DNA measurements, as well.

Participant selection and categorization in Figure 1 was firstly based on total HIV DNA and then integrated HIV DNA measurements were used to confirm assigned groups. We checked integrated DNA measurements to make sure that there were no participants that had total HIV DNA = 0 and integrated HIV DNA > 0 to explicitly ensure that a participant was not classified as “HIV DNA undetectable” due to assay detection failure. Thus, the actual integrated HIV DNA values were not used besides checking for 0 or >0 and were therefore not included as it makes no difference in the grouping when available (see below on the left panel). Please see further clarification in the methods section (line 454-455) and figure 1d is updated to include integrated HIV DNA comparisons.

Response Figure: Right panel: Integrated HIV DNA levels and left panel total HIV DNA categorized by week 48 reservoir measurements.

6. Please clarify what is being plotted on violin plots in Fig2D. Are these all cells from all participants, or are these individual values at one per participant? Text and callout to Fig. 2D-E on lines 167-8 suggest the latter, but then p values in Fig2D panel appear to be those of the single-cell (i.e., the former) approach. If plotting single cells, the total number of cells plotted should be noted. If one-per-participant values, then scatter plots of the 14 points would be preferable to violin plots.

Figure 2d violin plots represent gene expression of either IL1B or THBS1 from all single cells in the 14 total participants. Figure 2e (scatterplots) on the other hand is the average gene expression value per participant. We have updated the text to clarify this and added P values for Spearman correlation in Fig 2e (line 160-162).

7. Please include color codes in the figure or legend for Fig1A,B,C.

We have updated Figure 1a-c showing the labels in the figure.

8. Please provide a flow cytometry plot for latent and productive infection quantification for Fig5C.

Now included as supplementary Figure 7.

9. Please detail the methods to include amount of plasmids used for in vitro infection.

We have now included the amount of plasmid used in the in vitro infection. (lines 502-504)

10. What are the expression levels of NF-kB pathway genes in CD4+ T cell subsets from individuals with undetectable reservoir?

Supplementary Figure S6 shows the pathways (modules) found in CD4 memory cells (CD4 Tcm and Tem) and the pathway enrichment of modules associating with the reservoir phenotype in both cohorts. The top enriched pathway in the M4 module is the same Hallmark pathway also seen in CD14 monocytes, “TNF-alpha signaling via NF-kB”. Below we show violin plots in CD4 T cell subsets of the seven genes (DUSP2, EIF1, NR4A2, PER1, SLC2A3, TNFAIP3 and ZFP36) in the CD4 memory M4 module and the TNF-alpha signaling via NF-kB pathway. All genes but one were significantly higher in the undetectable group (p value and fold change) for both CD4 Tcm and Tem subsets. In comparison, we also show for this review CD4 Naïve T cells, which had lower expression of these genes. We now include these genes and their association with the undetectable phenotype in the results section to support our findings that individual genes in NF-kB pathways are enriched in CD4 T cells (lines 262-265). Findings for genes expressed in CD4 memory T cell subset from both ACTG and RV254 cohort is included in Supplementary Fig. 6f.

Response figure: Hub genes of the CD4 memory M4 module that are members of the TNF-alpha signaling via NF-kB pathway mostly have higher expression in the undetectable group than in the detectable group within CD4 memory subsets.

11. Line 162 states that “The DEGs that were most significant, with an average log fold change of >1. . . .” “Most significant” suggests magnitude of p value, but the sentence seems to intend to refer to fold change rather than p value. Would reword this.

Agree, we included the word ‘and’ to clarify.

Reviewer #2 (Remarks to the Author):

This study by Ehrenberg et al. is a small yet thoughtful investigation into host transcriptomic differences in acute/early-treated people with HIV (PWH) with “undetectable” versus “detectable” HIV reservoirs. To minimize potential confounders, the authors conducted their

analysis on a highly homogeneous primary cohort of 14 virally suppressed male Thai participants who initiated ART during Fiebig Stage III of acute HIV infection, using PBMC samples collected at 48 weeks of ART. They further validated their findings in a secondary cohort of 38 virally suppressed early-treated male U.S. PWH, again focusing on PBMC samples collected at 48 weeks of ART.

We thank the reviewer for recognizing our thoughtful investigation.

The findings align with a recent cohort-based study of virally suppressed PWH treated during early and chronic infection, which demonstrated an inverse association between IL-1b, TNF-a, NF-KB, and HIV reservoir size in peripheral CD4+ T cells. This suggests that the downregulation of host proinflammatory responses in bystander cells (e.g., monocytes and uninfected CD4+ T cells) may underlie this inverse association (citation 42). Given these parallels, it remains unclear whether the paper introduces a novel hypothesis or simply confirms these findings using single-cell sequencing data.

Citation 42 (revised citation 19) investigates associations of gene expression using bulk RNA-seq from peripheral CD4 T cells with different measures of HIV DNA or RNA largely from samples of treated HIV cohorts of chronic infection and some early treated cohorts. Their manuscript investigated gene expression only in CD4 T cells most likely because they are the major HIV reservoir. In contrast, we used an agnostic approach not based on prior knowledge of the reservoir to investigate the cell subsets (not just CD4 T cells) in peripheral blood that had the strongest association with the size of the reservoir. In the current study, samples were from treated AHI cohorts, which due to the timing of ART initiation are well characterized having longitudinal samples prior to treatment.

Response figure: Gene expression of IL1B in different single cell subsets.

This allowed us to choose samples without major confounders such as baseline VL and CD4 counts and consider HIV DNA levels prior to ART initiation for categorization. The distinctive nature of the cohorts and our screening of all major cell types in blood using scRNA-seq adds uniqueness to these findings. Findings were validated in two cohorts and also with another reservoir assay measuring intactness of provirus (IPDA). We further show that CD14+ monocytes have the strongest inverse association with the HIV DNA reservoir, which is new insight and also provides clarity to the previous finding since expression of genes such as IL1B is negligible in CD4 T cells.

Major Comments:

1. What was the basis for the decision to dichotomize the outcome variable for this small sample

size study (N=8 with "undetectable" vs. N=6 "detectable" HIV reservoir size by using total and integrated HIV DNA values)? This study design is problematic from a clinical and statistical standpoint. For example, for the former, the "undetectable" group might include wholly unique clinical phenotypes, such as elite controllers and/or post-treatment controllers (i.e., PWH able to control virus in the absence of therapy and PWH able to control virus after a period of time on ART, respectively). From a statistical perspective, these small comparator groups preclude regression modeling and inclusion of key clinical covariates that are predictive of HIV reservoir size (e.g., initial CD4+ T cell count, pre-ART viral load, etc.).

The basis to dichotomize reservoir size in the discovery cohort was to have the ability to look at extremes of the phenotype and carefully select participants to avoid confounders for gene expression analyses with phenotype of interest. Further these samples were followed from time of diagnosis at the acute timepoint, and provided valuable samples prior to ART initiation which allowed us to choose participants in each groups showing greatest differences in reservoir rather than time to treatment and other previously known factors such as pre-ART CD4 counts and VL (discussed in revised lines 82-83, 335-338, Figure 1d). That said, these findings were then validated in an independent cohort.

2. Because HIV total and/or integrated DNA (the primary endpoint/outcome) was not measurable in the (N=8) "undetectable" population, efforts to include these samples in any dose-response like analyses are also problematic. Specifically, the abundance of "tied undetectable values" from the lower end of the range may throw-off analyses such as linear or multiple regression (problems with leverage or weight) or even Spearman's correlation (ties complicate the calculation of exact p-values). The linear regression models should be replaced with (i) logistic regression models predicting if the outcome is detectable, (ii) a linear regression model conducted within the population with measurable HIV DNA reservoir values, or (iii) a more complicated model taking the censoring into account, like Tobit regression.

The basis of this analyses finding host factors associating with extremes of phenotypes, and in this case reservoir measurements. Thus, we cannot focus only on individuals with the detectable measures. Despite all the potential problems with "tied undetectable values", the linear regression model is appropriate due to its simplicity and feasibility, particularly given the small sample size, convergence issues with the logistic regression in the two-part model, the limited detectable data made the interaction analysis within measurable values infeasible and the complexity and reduced power of Tobit. We acknowledge the limitations of linear regression in addressing certain aspects of the data, but given the challenges with alternative models, this remains the most appropriate approach. To clarify further, we highlight in the methods why we chose linear regression analyses as it did not violate any assumptions (see revised methods, lines 659-664).

3. In Figure 2E, it appears that "average THBS1 expression" does not differ significantly between the two populations, whereas "average IL1B expression" may provide a more meaningful distinction. This observation suggests that THBS1, which does not replicate across cohorts, might be an artifact of the analysis method. Could the authors generate a similar figure to Figure 2E using IPDA data from the second cohort, where detection levels were not an issue? THBS1 analyses was not significant in the ACTG study not due to the assay for reservoir measurement, but because there was very low expression of this gene in this cohort. ACTG has sparse expression of THBS1 (5.6%), compared to 56% in RV254. IL1B expression in both

studies is shown for comparison. Population specific changes can impact responses to our findings and this information is now included in the results section and discussed (lines 220-223).

4. For the primary cohort, HIV reservoir was measured as integrated HIV DNA and total HIV DNA. For the validation cohort, HIV reservoir was measured as intact HIV DNA and defective HIV DNA. The authors should explain potential biological vs. artifactual reasons for why these assay results had different findings in the acute Thai vs. the ACTG (early-treated) cohorts? In both cohorts we consistently used the total HIV DNA measurement for analysis of IL1B gene expression with reservoir size (Fig 2e, Fig 3e). In the ACTG cohort, we were additionally able to analyze the intact/defective reservoir with another measurement called IPDA which is able to quantify subtype B, but not CRF01_AE.

5. Could the authors please clarify the rationale for performing TCR/BCR sequencing? Given that HIV-specific assays were not conducted, it would be helpful to understand how global TCR and/or BCR differences are relevant to the study. This aspect was not clearly explained in the Methods or Results sections. If these findings were included solely as a feature of the 5' 10X Genomics kit, the authors might consider revisiting whether their inclusion is necessary. Data from different multiomics modalities was included solely for the purposes of immune profiling, and to satisfy the unbiased nature of the analyses. We hope making such data public would benefit the research community.

6. Could the authors elaborate on the decision to include only acute and early-treated PWH in the study design? Additionally, how might the findings differ, if at all, in chronic-treated PWH? The reason why we focused our analyses on acute treated cohorts was to take into consideration major confounders of gene expression due to differences in HIV-1 seroconversion, such as timing of infection which is not possible in chronic infection. But as mentioned in the discussion section (lines 402-403), we agree that exploration in chronic treated cohorts is worthwhile.

7. Please clarify the statement in the Methods section, “Assessment of model diagnostics [...] showed that the assumptions of the linear models were reasonable after removing one outlier.” How was the outlier identified? Which participant is represented by the outlier data? Besides improving the model fit, is there reason to believe this sample should be omitted? Did removing this sample affect the direction or significance of the results? Model diagnostics were carried out using the `gvlma()` function in R and we evaluated diagnostic plots to ensure that model assumptions were met. The outlier was identified using Cook's distance which showed that sample (detectable_6) was an influential point and including this participant data violates the assumptions and cannot be run. We have updated the methods section to address this (lines 659-664).

8. The introduction could benefit from a more comprehensive explanation of how the findings from the current study build upon or support the existing literature. While it is noted that these findings are primarily observed in monocytes—already recognized as the primary source of IL-1b in peripheral blood—this point alone may not sufficiently contextualize the study's contribution. It might be helpful to revise this section to incorporate key references, such as

citation 42, and to explicitly describe how the current study advances or complements these prior studies.

We have previously mentioned citation 42 (revised 19) in the discussion, but now also include in the introduction.

9. The study is well-written and presents valuable findings; however, it could benefit from a more in-depth discussion of the clinical and translational implications of these results. Could the authors elaborate on their perspective regarding the observed inverse association between proinflammatory pathways and HIV reservoir size?

Direct clinical and translation implications of these findings remain to be investigated, but our findings suggests that agents activating NF-kB may work in the long-term. We have updated the discussion to reflect this perspective though cautiously. These findings support that immune cells other than T cells can modulate the HIV reservoir in a clinical cohort, and that this effect may be influenced by specific genes and pathways via NF-kB signaling (see lines 414-427). This inverse correlation also speaks to innate differences in people perhaps even prior to HIV infection that control reservoir size which needs to be explored. See response to reviewer 1 point 2.

Minor Comments:

1. Overall, the directionality of the associations could be made clearer, as it is somewhat challenging to follow. Using more explicit terms such as “increase” or “decrease” (e.g., in host gene expression in relation to HIV reservoir) rather than “differential” or “change” would enhance clarity and help readers better understand the relationships being described, as they are not immediately intuitive.

IL1B gene expression in monocytes has an inverse correlation with reservoir size. Directionality is first mentioned in lines (102-103), the figures clearly illustrate direction (Fig. 2e, 3e) and one of the reasons for including a monotonic line to clarify negative correlation of IL1B genes expression with reservoir size (see later response).

2. A model or summary figure could be very helpful in illustrating the directionality of the observed associations. It would also enhance the understanding of the biological relevance of the results.

We agree and have included a figure in the supplementary section (S12) which provides a working model of our findings. We hypothesize that IL1B may affect the latent HIV reservoir by 1) acting as a natural LRA, 2) contributing to reduced seeding of the reservoir, and 3) changing the composition of CD4+ T cell subsets.

3. In Figure 2E (and 3E), the authors should remove the linear trend line, as it suggests a linear correlation. Since the figure appears to present a Spearman correlation, which does not necessarily imply a linear relationship, adjusting this could provide a more accurate representation.

The lines were mainly to show directionality of the association and not establish linearity. We have now replaced the linear trend line with a spline to highlight directionality of points while avoiding misinterpretation (Figures 2e, 3e). The goal is to highlight that this is a monotonic relationship (not necessarily linear). The legends have been updated accordingly.

4. It seems that there may be some missing column headings in Extended Table 1. Specifically,

the labels for the columns “(copies/mL) (AHI)”, “(cells/mm³) (AHI)”, “(copies/mL) (ART)”, and “(cells/mm³) (ART)” are unclear.

We thank the reviewer for pointing this issue that happened during excel to pdf conversion. We have updated the height of the header row so that relevant information is no longer missing. The column labels in question should read:

HIV-1 RNA (copies/mL) (AHI)	CD4 counts (cells/mm ³) (AHI)	HIV-1 RNA (copies/mL) (ART)	CD4 counts (cells/mm ³) (ART)
--	-----------------------------------	--

Reviewer #3 (Remarks to the Author):

In this manuscript, the authors illustrated the transcriptomic differences between PBMCs from patients with varying HIV DNA loads. They highlighted IL1B in CD14+ monocyte as a potential mediator of HIV reservoirs via the NF-kb signaling in CD4+ T cells. The manuscript is interesting and well-written. The analyses performed are rigorous, however require some methodological details. Additionally, to strengthen the findings further, I have some suggestions to fill the missing puzzles of the story.

We thank the reviewer for acknowledging the rigor of the study and their suggestions.

Major comments:

1. Fig. S1 was far not adequate to support the cell type annotation in Fig. 2A and Fig. 3C. A full up-regulated gene list for each cell subset was required for both Fig. 2A and Fig. 3C. Additionally, the resolution of Fig. S1 is too low to match it with the cell identifications in Fig. 3C.

Fig S1 resolution has been increased, additional 9 feature markers have been included and the colors have been adjusted for clarity. The markers used for annotation are standard and are rarely provided in papers as a table, but we include a link to our in-house annotation repository on github for reference and included this link in the methods section(lines 588-589) (https://github.com/thomaslab-MHRP/scRNA-seq_annotation_resources/blob/main/GEX_markers_for_cell_annotation.md).

2. To support the authors’ hypothesis on line 263 about the IL1B-IL1R signaling, it would be useful to perform a cell-cell interaction analysis to confirm the potential interaction from CD14+ monocyte: IL1B to CD4+ T cell: IL1R in the single-cell data.

We checked cell-cell interaction using a software called CellChat v2.1.2 (PMID: 39289562) which specifically looks for interactions between ligand in a cell type and receptors in another cell type. We checked for IL1B in CD14 monocytes and IL1R in CD4+ memory T cells and did not find an interaction between these cell types. We think this is because the three receptors for IL1B listed in the database (IL1RAP, IL1R1 and IL1R2) have sparse expression (few cells) in CD4+ memory T cells. This pattern was the same in the ACTG study also, and so we did not find meaningful interaction using CellChat and its existing databases for these ligand-receptor interactions. We now include in the results section (lines 255-258).

3. It is necessary to show whether NF-kb activated differently in CD4 T cells from undetectable and detectable groups in scRNAseq data to support the hypothesis on lines 265-266.

We now include these genes and their association with the undetectable phenotype in the results section to support our findings that individual genes in NF- κ B pathways are enriched in CD4 T cells (lines 262-265). Findings for genes expressed in CD4 memory T cell subset from both ACTG and RV254 cohort is included in Supplementary 6. Please also see response to reviewer 1 (minor point 10).

Minor comments:

1. As shown in Fig. 1C, it seems that all patients of blue groups have lower HIV DNA copies than patients of red groups at week 0, therefore, I am a little surprised by the non-significant difference indicated in Fig. 1D.

The groups are significantly different in Figure 1c which we now include P values. There is also a significant difference between the undetectable and detectable groups at week 0 (AHI) ($p = 0.002$), which is now included in the same table in Fig 1d.

2. Systematic comparisons/integration were suggested for the two scRNAseq datasets to show the consistence beyond only number of DEGs and IL1B. For example, a scatter plot showing the changes of the same genes in the two datasets to support the consistent alterations in CD14+ monocytes.

Per the reviewer's suggestion, we investigated global changes in monocytes that were nominally significant and overlapping in directionality between the two studies and found many genes that associated with reservoir size, including IL1B. Unlike the method we used, this method is less stringent and found >250 genes for focused downstream analyses to investigate functional effects. Our method used a more agnostic approach to identify gene expression patterns with coordinated expression affecting reservoir size using WGCNA as shown in Figure 4. Again, the NF- κ B pathway is still the top hit using this method which supports our findings.

Response figure: Left: Scatter plot of DEGs in CD14 monocytes comparing the absolute logFC from RV254 with the absolute coefficient from MAST analysis in ACTG for all genes that are nominally significant in both studies having the same directionality (color of the dot). Genes that were significant (p value adj < 0.05) in both studies are labeled. Right: Pathway analyses of the genes in the scatter plot.

Reviewer #1 (Remarks to the Author):

Responses are provided in red font.

Several concerns about the initial submission have now been addressed in the revision and rebuttal, but there are two major concerns about the revision.

1. The section of the Results entitled “Monocyte-expressed genes in conjunction with central memory CD4+ T cell frequencies were associated with decreased reservoir size” is very unclear.

a. Line 176 reports that there were 117 cell populations identified by flow cytometry. This sounds complicated. However, this reviewer does not see a list of the 117 cell populations (i.e., what are their markers?), nor is there any technical or conceptual explanation of the analysis process that defined them. Methods state only that “Data were analyzed with FlowJo v.9.9.6 or higher (Becton Dickinson).”

A total of 117 cell populations were identified and annotated by cell surface marker expression, including monocytes, DC, NK and B cell subsets as well as CD4+/CD8+ T cell activation, exhaustion and memory status from PBMC isolated at the same time as those used in scRNA-seq analyses. We have now included a new supplementary data file 2 defining the marker gating used for all 117 populations and updated the results (lines 178-181) and methods (line 506) section.

b. In response to the question from the initial submission about CD39, which was the only marker associated with HIV reservoir size in univariate analysis, the revision includes the following sentence: “These findings agree with earlier findings of the role of CD39 expression in HIV pathogenesis and latency (refs 26-28).” This reviewer was unaware of the role of CD39 expression in HIV pathogenesis and latency. Among the cited references, the first is a review article, the second is a brief phenotyping study providing little mechanistic insight, and the third is a more detailed study that is not unpacked at all here. As the only hit in a univariate analysis, the CD39 finding would seem to deserve a more thorough thought process.

There was an error in the references for CD39 in the manuscript versus the citations provided in the previous response and we apologize for this mistake. We have now corrected this to cite 4 primary articles that support a role of CD39 with increased HIV DNA levels, latency, immune activation and disease progression (PMID: 31681335, 35529864, 37910505, 21750674) in the results (lines 183-185) and discussion (lines 367-369) sections.

c. Although the rebuttal letter and Methods provide some technical explanation of how the “multiple regression model with two-way interaction” analysis was performed, this reviewer remains very confused about why this type of analysis was selected, the logic of its design, and exactly what the findings are (measured or modeled). The Discussion should offer a detailed mechanistic explanation of the findings that will unify them for the reader, while acknowledging ways in which this explanation could be wrong. The Discussion section in the revision falls short of this standard.

The exploratory two-way interaction with multiple regression was designed to investigate if there was a link between our monocyte gene expression association with reservoir size and the cell population frequency data generated using flow cytometry. A standard main-effects regression assumes that each variable influences the outcome independently, which was not our focus.

Hence our choice of analysis was based on a multivariate model including an interaction term, that we hypothesized could identify immune populations that influence our primary finding of monocyte gene expression associating with reservoir size. We used measures of monocyte gene expression and the frequency of immune populations to model the reservoir size. We found from a total of 117 cell populations, that only frequency of a subset of CD4+ T cells showed a significant interaction with monocyte gene expression and affects reservoir size. As CD4+ T cells are known to harbor HIV this was consistent with our single-cell RNA-seq findings from monocytes indicating an effect on the latent reservoir. We now update results (lines 183-188, 211-213) to clarify better the rationale for the multivariate regression analyses in the results and the section title, and discussion (lines 368-375) to include this interpretation of the implications. This complements our previous discussion sections that address the subsequent mechanistic insight gained from the in vitro experiments (lines 376-406), supplementary figure 12 which unifies these points schematically, and limitations of this study (lines 407-424).

2. Flow cytometry in Supplementary Fig. 7 and related results in Fig. 5 appear problematic. Supplementary Fig. 7 does not show data from mock-infected controls. Therefore, it is difficult to be sure how much of each small cell population is background signal and how much is real. Considering this, it is difficult to interpret small absolute changes in frequencies of latently-infected and actively-infected cells in this model.

To address this concern, we now show the mock control in the revised supplementary figure 7 and subtracted the background in the revised Figure 5c. The background was generally low and subtracting it did not affect the significance of the findings or our conclusions.

Additional minor comments are as follows.

1. This reviewer did not understand the meaning of Lines 164-166, which appear to report an association between THSB1 and IL1B expression in CD14+ monocytes and rate of reservoir decay. This reviewer does not see which portions of Supplementary Figs 4c and 4d refer to reservoir decay. This needs to be clarified.

In supplementary Figure 4c-d we performed a MAST analysis using reservoir decay between weeks 0 and 48 as a continuous outcome variable. The two supplementary figures show that CD14+ monocytes have the strongest association with reservoir decay (Supplementary Figure 4c), and *THBS1* and *IL1B* are similarly significant in this analysis in CD14+ monocytes (Supplementary Figure 4d) Thus, in addition to reservoir size at week 48, reservoir decay also showed similar associations. We have updated lines 164-170 in the results section to clarify better.

2. In Fig 1c, the y-axis is not labeled. Also, the uppermost blue dot at Week 0 does not have a segment connecting to week 48, while the second uppermost blue dot appears to have two segments.

Figures 1b and 1c share the same y-axis (Total HIV DNA copies/ 10^6 cells). We have updated the y-axes for 1b and 1c to clarify the shared axis. We have also reduced the amount of jittering in 1c so that the lines can be more easily matched with their points to avoid the apparent disconnect between some of them.

3. In Supplementary Fig. 4, please clarify the relationship between participants shown in panels a

and b. Presumably, the 5 participants shown as detectable and 6 shown as undetectable in panel b were assigned to the same groups in panel a, i.e., reservoir detectability agreed between measurements made in CD4+ T cells and those made in PBMC. Assuming this is true, it would be clearer to combine all data in one graph. If not true, the discrepancy should be explained.

This is correct. There is no discrepancy in categorization when using HIV DNA measurements from PBMC or from CD4 T cells. To make it clearer, we have combined previous S4a and b into one graph. Vertical arrows indicate participants for whom we only have PBMC reservoir measurements and not CD4 T cell measurements. The figure legend has been updated accordingly.

4. What is the meaning of the “-“ character in Supplementary table 1, which reports frequencies of different cell populations in RV254 and ACTG cohorts?

The “-” indicates populations that did not have an exact equivalent in one of the two cohorts. For example, in RV254 we found separate CD4+ T_{EM} and T_{CM} clusters, but in ACTG our CD4+ memory T cells instead split into a larger CD4 memory and a smaller cluster with interferon upregulated genes. Although majority of the clusters were identical, because the method was unsupervised clustering and not label transfer there were minor population differences between the two cohorts which did not affect results or conclusions in this study. Please refer to lines 606-609 in the methods section.

5. Lines 219-220 state that while HIV DNA levels are correlated with IL1B expression in monocytes, they are not correlated with THBS1 expression in monocytes. Fig 3e should show the THBS1 data to support this statement.

Below is the equivalent figure to 3E using *THBS1*. The p values from a Spearman correlation of all samples and Mann Whitney U test comparing the two extreme phenotype groups are both ≥ 0.05 and not significant (see below). We prefer not to include the figure below, but now include in the text the rho and P values (lines 235-236).

6. If the authors consider it to be analytically appropriate, the manuscript might mention whether there were any genes that were differentially expressed between detectable and undetectable groups in the ACTG validation cohort but not the RV254 exploratory cohort.

Since ACTG was the secondary and validation cohort it would be inappropriate to analyze in a reverse fashion for this study.

7. Supplementary Fig 6a-6e are not mentioned in the manuscript text. All included display items should be called out in the text.

We have updated this oversight (lines 267, 270).

Reviewer #2 (Remarks to the Author):

The authors have done a nice job of addressing my comments. I have no further comment

Reviewer #3 (Remarks to the Author):

My concerns have been acceptably addressed.

Reviewer #3 (Remarks on code availability):

The codes were well-documented and enough for reproducing the major findings.

Reviewer #4 (Remarks to the Author):

Reviewer #5 (Remarks to the Author):

REVIEWERS' COMMENTS

Reviewer #1 (Remarks to the Author):

The authors have addressed this reviewer's concerns. The following minor comments can be addressed editorially before publication.

Results section 3: Initially, the rationale for conducting the multivariate regression analysis was unclear, but the explanation provided has been helpful. Regarding the methodology/procedures, any further clarification for readers not expert in these statistical methods could be beneficial.

We are pleased that the rationale for this analysis is now clear. The method for the multivariate linear regression using an interaction term is described in the methods section with detailed explanation of the predictor and response variables, and all related attributes used in the regression analysis. We now also include a reference aimed at non-experts interested in the theoretical basis of this analysis.

Results section 3: The authors now refer to a "subset of memory T cells" instead of "central memory T cell subsets." This reviewer would like to clarify that previous comments aimed to understand the rationale behind the experiment, and were based on the sense that a link to central memory was insufficiently explained. From this reviewer's perspective, it is appropriate to refer to central memory in the manuscript. Any further speculative explanations that might be offered to help readers imagine what the mechanism(s) of the link to central memory might be would strengthen the manuscript.

We have updated to include the description of this specific subset as central memory T cells. There is prior data showing that IL1B acts as a "licensing signal" for memory CD4 T cells, enabling them to produce effector cytokines and participate in immune responses (PMID: 30093707). It is possible that in our finding this might be more relevant to central memory CD4 T cells, but more work is needed to establish if this interaction is specific to this subset rather than other CD4 cell types. We cautiously mention this possibility in the discussion section and include this reference.